# Adversarial Robustness in Graph Neural Networks: A Hamiltonian Approach

**Kai Zhao**[*]
Nanyang Technological University

**Qiyu Kang**[*†]
Nanyang Technological University

**Yang Song**[*]
C3 AI, Singapore

**Rui She**
Nanyang Technological University

**Sijie Wang**
Nanyang Technological University

**Wee Peng Tay**
Nanyang Technological University

## Abstract

Graph neural networks (GNNs) are vulnerable to adversarial perturbations, including those that affect both node features and graph topology. This paper investigates GNNs derived from diverse neural flows, concentrating on their connection to various stability notions such as BIBO stability, Lyapunov stability, structural stability, and conservative stability. We argue that Lyapunov stability, despite its common use, does not necessarily ensure adversarial robustness. Inspired by physics principles, we advocate for the use of conservative Hamiltonian neural flows to construct GNNs that are robust to adversarial attacks. The adversarial robustness of different neural flow GNNs is empirically compared on several benchmark datasets under a variety of adversarial attacks. Extensive numerical experiments demonstrate that GNNs leveraging conservative Hamiltonian flows with Lyapunov stability substantially improve robustness against adversarial perturbations. The implementation code of experiments is available at `https://github.com/zknus/NeurIPS-2023-HANG-Robustness`.

## 1 Introduction

Graph neural networks (GNNs) [1–8] have achieved great success in inference tasks involving graph-structured data, including applications from social media networks, molecular chemistry, and mobility networks. However, GNNs are known to be vulnerable to adversarial attacks [9]. To fool a trained GNN, adversaries can either add new nodes to the graph during the inference phase or remove/add edges from/to the graph. The former is called an injection attack [10–13], and the latter is called a modification attack [14–16]. In some works [9, 17], node feature perturbations are also considered to enable stronger modification attacks.

Neural ordinary differential equation (ODE) networks [18] have recently gained popularity due to their inherent robustness [19–24]. Neural ODEs can be considered as a continuous analog of ResNet [25]. Many neural ODE networks have since been proposed, including but not limited to [19, 20, 26–28]. Using neural ODEs, we can constrain the input and output of a neural network to follow certain physics laws. Injecting physics constraints to black-box neural networks improves neural networks' explainability. More recently, neural ODEs have also been successfully applied to GNNs by modeling

---

[*]First three authors contributed equally to this work.

[†]Correspondence to: Qiyu Kang <kang0080@e.ntu.edu.sg>

the way nodes exchange information given the adjacency structure of the underlying graph. We call them *graph neural flows* which enables the interpretation of GNNs as evolutionary dynamical systems. These system equations can be learned by instantiating them using neural ODEs [29–34]. For instance, [29, 30] models the message-passing process, i.e., feature exchanges between nodes, as the heat diffusion, while [31, 32] model the message passing process as the Beltrami diffusion. The reference [35] models the graph nodes as coupled oscillators with a coupled oscillating ODE guiding the message-passing process.

Although adversarial robustness of GNNs has been investigated in various works, including [9, 13, 17], robustness study of graph neural flows is still in its fancy. To the best of our knowledge, only the recent paper [32] has started to formulate theoretical insights into why graph neural diffusion is generally more robust against topology perturbation than conventional GNNs. The concept of Lyapunov stability was used in [20, 21]. However, there are many different notions of stability in the dynamic system literature [36, 37]. In this paper, we focus on the study of different notions of stability for graph neural flows and investigate which notion is most strongly connected to adversarial robustness. We impose an energy conservation constraint on graph neural flows that lead to a Hamiltonian graph neural flow. We find that energy-conservative graph Hamiltonian flows endowed with Lyapunov stability improve the robustness the most as compared to other existing stable graph neural flows.

**Main contributions.** This research is centered on examining various stability notions within the realm of graph neural flows, especially as they relate to adversarial robustness. Our main contributions are summarized as follows:

1. We revisit the definitions of stability from the perspective of dynamical systems as applicable to graph neural flows. We argue that vanilla Lyapunov stability does not necessarily confer adversarial robustness and provide a rationale for this observation.

2. We propose Hamiltonian-inspired graph neural ODEs, noted for their energy-conservative nature. We perform comprehensive numerical experiments to verify their performance on standard benchmark datasets. Crucially, our results demonstrate that Hamiltonian flow GNNs present enhanced robustness against various adversarial perturbations. Moreover, it is found that the effectiveness of Lyapunov stability becomes pronounced when layered on top of Hamiltonian flow GNNs, thereby fortifying their adversarial robustness.

The rest of this paper is organized as follows. We introduce various stability notions from a dynamical system perspective in Section 2. A review of existing graph neural flows is presented in Section 3 with links to the stability notions defined in Section 2. We present a new type of graph neural flow inspired by the Hamiltonian system with energy conservation in Section 4. Two different variants of this model are proposed in Section 5. Section 6 details our extensive experimental outcomes. The supplementary section provides an overview of related studies, an exhaustive outline of the algorithm, further insights into model robustness, supplementary experimental data, and the proofs for the theoretical propositions made throughout the paper.

## 2    Stability in Dynamical Systems

It is well known that a small perturbation at the input of an unstable dynamical system will result in a large distortion in the system's output. In this section, we first introduce various types of stability in dynamical physical systems and then relate them to graph neural flows. We consider the evolution of a dynamical system that is described as the following autonomous nonlinear differential equation:

$$\frac{\mathrm{d}\mathbf{z}(t)}{\mathrm{d}t} = f_{\boldsymbol{\theta}}(\mathbf{z}(t)), \tag{1}$$

where $f_{\boldsymbol{\theta}} : \mathbb{R}^n \to \mathbb{R}^n$ denotes the system dynamics, which may be non-linear in general, $\boldsymbol{\theta}$ denotes the system parameters, and $\mathbf{z} : [0, \infty) \to \mathbb{R}^n$ represents the $n$-dimensional system state.

We first introduce the stability notions from a dynamical systems perspective that is related to the GNN robustness against *node feature perturbation*.

**Definition 1** (BIBO stability)**.** *The system is called BIBO (bounded input bounded output) stable if for any bounded input, there exists a constant $M$ s.t. the output $\|\mathbf{z}(t)\| < M, \forall t \geq 0$.*

Suppose $f$ has an equilibrium at $\mathbf{z}_e$ so that $f_{\boldsymbol{\theta}}(\mathbf{z}_e) = 0$. We can define the stability notion for $\mathbf{z}_e$.

**Definition 2** (Lyapunov stability and asymptotically stable [38]). *The equilibrium $\mathbf{z}_e$ is Lyapunov stable if for every $\epsilon > 0$, there exists a $\delta > 0$ such that, if $\|\mathbf{z}(0) - \mathbf{z}_e\| < \delta$, then for every $t \geq 0$ we have $\|\mathbf{z}(t) - \mathbf{z}_e\| < \epsilon$. Furthermore, the equilibrium point $\mathbf{z}_e$ is said to be asymptotically stable if it is Lyapunov stable and there exists a $\delta' > 0$ such that if $\|\mathbf{z}(0) - \mathbf{z}_e\| < \delta'$, then $\lim_{t\to\infty} \|\mathbf{z}(t) - \mathbf{z}_e\| = 0$.*

**Remark 1.** *Lyapunov stability indicates that the solutions whose initial points are near an equilibrium point $\mathbf{z}_e$ stay near $\mathbf{z}_e$ forever. For the special linear time-invariant system $\mathrm{d}\mathbf{z}(t)/\mathrm{d}t = \mathbf{A}\mathbf{z}(t)$ with a constant matrix $\mathbf{A}$, it is Lyapunov stable if and only if all eigenvalues of $\mathbf{A}$ have non-positive real parts and those with zero real parts are the simple roots of the minimal polynomial of $\mathbf{A}$ [39, 40]. Asymptotically stable means that not only do trajectories stay near $\mathbf{z}_e$ for all time (Lyapunov stability), but trajectories also converge to $\mathbf{z}_e$ as time goes to infinity (asymptotic stability).*

We next introduce the concept of structural stability from dynamical systems theory, which is related to the robustness of GNNs against *graph topological perturbation*. It describes the sensitivity of the qualitative features of a solution to changes in parameters $\boldsymbol{\theta}$. The definition of structural stability requires the introduction of a topology on the space of $\mathbf{z}$ in (1), which we do not however present here rigorously due to space constraints and not to distract the reader with too much mathematical details. Instead, we provide a qualitative description of structural stability to elucidate how it can indicate the robustness of a graph neural flow against topology perturbations.

**Definition 3** (Structural stability). *Unlike Lyapunov stability, which considers perturbations of initial conditions for a fixed $f_{\boldsymbol{\theta}}$, structural stability deals with perturbations of the dynamic function $f_{\boldsymbol{\theta}}$ by perturbing the parameter $\boldsymbol{\theta}$. The qualitative behavior of the solution is unaffected by small perturbations of $f_{\boldsymbol{\theta}}$ in the sense that there is a homeomorphism that globally maps the original solution to the solution under perturbation.*

**Remark 2.** *In the graph neural flows to be detailed in Section 3 and Section 4, the parameter $\boldsymbol{\theta}$ includes the graph topology (i.e., the adjacency matrix) and learnable neural network weights. Unlike adversarial attacks on other deep learning neural networks where the attacker targets only the input $\mathbf{z}$, it is worth noting that adversaries for GNNs can also attack the graph topology, which forms part of $\boldsymbol{\theta}$. If there are different Lyapunov stable equilibrium points, one for each class of nodes, one intuitive example of breaking structural stability in graph neural flows is by perturbing the graph topology in such a way that there are strictly fewer equilibrium points than the number of classes.*

In this study, we will propose GNNs drawing inspiration from Hamiltonian mechanics. In a Hamiltonian system, $\mathbf{z} = (q, p) \in \mathbb{R}^{2n}$ refers to the generalized coordinates, with $q$ and $p$ corresponding to the generalized position and momentum, respectively. The dynamical system is characterized by the following nonlinear differential equation:

$$\frac{\mathrm{d}\mathbf{z}(t)}{\mathrm{d}t} = J\nabla H(\mathbf{z}(t)), \tag{2}$$

where $\nabla H(\mathbf{z})$ is the gradient of a scalar function $H$ at $\mathbf{z}$ and $J = \begin{pmatrix} 0 & I \\ -I & 0 \end{pmatrix}$ is the $2n \times 2n$ skew-symmetric matrix with $I$ being the $n \times n$ identity matrix.

We now turn our attention to the notion of conservative stability in dynamical systems. It is worth noting that a general dynamical system, as characterized in (1), might not consistently resonate with traditional perspectives on energy and conservation, especially when compared to physics-inspired neural networks, like (2).

**Definition 4** (Conservative stability). *In a dynamical system inspired by physical principles, such as (2), a conserved quantity might be present. This quantity, which frequently embodies the notion of the system's energy, remains invariant along the system's evolution trajectory $\mathbf{z}(t)$.*

Our focus in this work is on graph neural flows that can be described by either (1) or (2). Chamberlain et al. [29] postulate that many GNN architectures such as GAT can be construed as discrete versions of (1) via different choices of the function $f_{\boldsymbol{\theta}}$ and discretization schemes. Therefore, the stability definitions provided above can offer additional insights into many popular GNNs. Most existing graph neural flows only scrutinize the BIBO/Lyapunov stability of their system. For instance, GRAND [29] proposes BIBO/Lyapunov stability against node feature perturbation over $\mathbf{z}$. However, the more fundamental structural stability in graph neural flows, which is related to the robustness against graph topological changes, remains largely unexplored. Some models, such as GraphCON [29, 35], exhibit conservative stability under certain conditions. We direct the reader to Section 3 and Table 1 for a comprehensive discussion of the stability properties of each model.

# 3 Existing Graph Neural Flows and Stability

Consider an undirected, weighted graph $\mathcal{G} = (\mathcal{V}, \mathcal{E})$ where $\mathcal{V}$ is a finite set of vertices and $\mathcal{E} \subset \mathcal{V} \times \mathcal{V}$ denotes the set of edges. The adjacency matrix of the graph is denoted as $(\mathbf{W}[u, v]) = \mathbf{W}([v, u])$ for all $[u, v] \in \mathcal{E}$. Let $\mathbf{X}(t) \in \mathbb{R}^{|\mathcal{V}| \times r}$ represent the features associated with the vertices at time $t$. The feature vector for the $i$-th node in $\mathcal{V}$ at time $t$ can be represented as the $i$-th row of $\mathbf{X}(t)$, indicated by $\mathbf{x}_i^\mathsf{T}(t)$. In this section, we introduce several graph neural flows on $\mathcal{G}$, categorizing them according to the stability concepts outlined in Section 2.

**GRAND:** Inspired by the heat diffusion equation, GRAND [29] employs the following dynamical system:

$$\frac{\mathrm{d}\mathbf{X}(t)}{\mathrm{d}t} = \overline{\mathbf{A}}_G(\mathbf{X}(t))\mathbf{X}(t) := (\mathbf{A}_G(\mathbf{X}(t)) - \alpha\mathbf{I})\mathbf{X}(t), \tag{3}$$

with the initial condition $\mathbf{X}(0)$. Within this model, $\mathbf{A}_G(\mathbf{X}(t))$ is either a time-invariant static matrix, represented as GRAND-l, or a trainable time-variant attention matrix $(a_G(\mathbf{x}_i(t), \mathbf{x}_j(t)))$, labeled as GRAND-nl, reflecting the graph's evolutionary features. The function $a_G(\cdot)$ calculates similarity for pairs of vertices, and $\mathbf{I}$ is an identity matrix with dimensions that fit the context. In [29], $\alpha$ is set to be 1. Let $\mathbf{D}$ be the diagonal node degree matrix where $\mathbf{D}[u, u] = \sum_{v \in \mathcal{V}} \mathbf{W}[u, v]$.

**Theorem 1.** *We can prove the following stability:*

1). *For GRAND-nl, if the attention matrix $\mathbf{A}_G(\mathbf{X}(t))$ is set as a doubly stochastic attention [41], we have BIBO stability and Lyapunov stability for any $\alpha \geq 1$. When $\alpha > 1$, it reaches global asymptotic stability under any perturbation.*
2). *Within the GRAND-l setting, if $\mathbf{A}_G$ is set as a constant column- or row-stochastic matrix, such as the normalized adjacency matrices $\mathbf{WD}^{-1}$ or $\mathbf{D}^{-1}\mathbf{W}$, global asymptotic stability is achieved for $\alpha > 1$ under any perturbation. If the graph is additionally assumed to be strongly connected [42][Sec.6.3], BIBO and Lyapunov stability are realized for $\alpha = 1$.*
3). *Furthermore, when $\mathbf{A}_G$ is specifically a constant column-stochastic matrix like $\mathbf{WD}^{-1}$ and $\alpha = 1$, GRAND conserves a quantity that can be interpreted as energy. Furthermore, in this settting, asymptotic stability is attained when the graph is aperiodic and strongly connected and the perturbations on $\mathbf{X}(0)$ ensure unaltered column summations.*

**BLEND:** *In comparison to GRAND, BLEND [31] introduces the use of positional encodings. Following a similar line of reasoning to that used for GRAND, BLEND also exhibits BIBO/Lyapunov stability as stated in Theorem 1. Moreover, it is noteworthy that if positional features $\mathbf{U}(t)$ are eliminated, for instance by setting them as a constant, BLEND simplifies to the GRAND model.*

**GraphCON:** Inspired by oscillator dynamical systems, GraphCON is a graph neural flow proposed in [35] and defined as

$$\begin{cases} \frac{\mathrm{d}\mathbf{Y}(t)}{\mathrm{d}t} = \sigma(\mathbf{F}_\theta(\mathbf{X}(t), t)) - \gamma\mathbf{X}(t) - \alpha\mathbf{Y}(t), \\ \frac{\mathrm{d}\mathbf{X}(t)}{\mathrm{d}t} = \mathbf{Y}(t), \end{cases} \tag{4}$$

where $\mathbf{F}_\theta(\cdot)$ is a learnable 1-neighborhood coupling function, $\mathbf{Y}(t)$ is an auxiliary velocity variable, $\sigma$ denotes an activation function, and $\gamma$ and $\alpha$ are tunable parameters.

*As described in [35, Proposition 3.1], under specific settings where $\sigma$ is the identity function and $\mathbf{F}_\theta(\mathbf{X}(t), t) = \mathbf{AX}(t)$ with $\mathbf{A}$ being a constant matrix, GraphCON conserves Dirichlet energy* (11)*, thereby demonstrating conservative stability.*

**GraphBel:** Generalizing the Beltrami flow, mean curvature flow and heat flow, a stable graph neural flow [32] is designed as

$$\frac{\mathrm{d}\mathbf{X}(t)}{\mathrm{d}t} = (\mathbf{A_S}(\mathbf{X}(t)) \odot \mathbf{B_S}(\mathbf{X}(t)) - \Psi(\mathbf{X}(t)))\mathbf{X}(t), \tag{5}$$

where $\odot$ is the element-wise multiplication. $\mathbf{A_S}(\cdot)$ and $\mathbf{B_S}(\cdot)$ are learnable attention function and normalized vector map, respectively. $\Psi(\mathbf{X}(t))$ is a diagonal matrix in which $\Psi(\mathbf{x}_i, \mathbf{x}_i) = \sum_{\mathbf{x}_j}(\mathbf{A} \odot \mathbf{B})(\mathbf{x}_i, \mathbf{x}_j)$.

*Analogous to BLEND, under certain conditions with $\Psi(\mathbf{X}(t)) = \mathbf{B_S}(\mathbf{X}(t)) = \mathbf{I}$, GraphBel simplifies to the GRAND model. Consequently, it exhibits BIBO/Lyapunov stability in certain scenarios.*

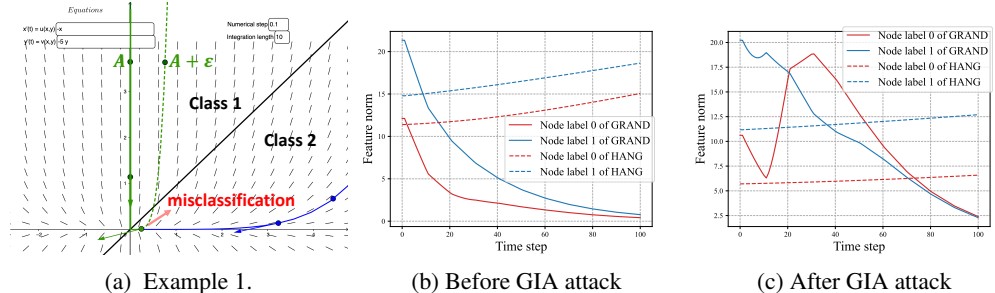

| (a) Example 1. | (b) Before GIA attack | (c) After GIA attack |

Figure 1: (a): We plot the vector field and the system solution trajectories of Example 1. (b) and (c): In GRAND, the node features' energy tends to converge towards each other. In HANG, we observe the node features' energy remains relatively stable over time. Two nodes are from different classes.

The incorporation of ODEs via graph neural flows may enhance the stability of graph feature representations. A summarized relationship between model stability and these graph neural flows can be found in Table 1.

## 3.1 Lyapunov Stability vs. Node Classification Robustness:

At first glance, Lyapunov stability has a strong correlation with node classification robustness against feature perturbations. However, before diving into experimental evidence, we point out an important conclusion: **Lyapunov stability *by itself* does not necessarily imply adversarial robustness.** Consider a scenario where a graph neural flow has only one equilibrium point $\mathbf{z}_e$, while the node features are derived from more than one class. In a Lyapunov asymptotically stable graph neural flow, such as GRAND (as shown in Theorem 1), all node features across different classes would inevitably converge to a single point $\mathbf{z}_e$ due to global contraction. We note that this is why the model in [20] requires a diversity-promoting layer to ensure that different classes converge to different Lyapunov-stable equilibrium points.

**Example 1.** *We provide an example to demonstrate our claim. Consider the following Lyapunov stable ODE*

$$\dot{\mathbf{x}}(t) = \begin{pmatrix} -1 & 0 \\ 0 & -5 \end{pmatrix} \mathbf{x}(t) \tag{6}$$

*with initial condition $\mathbf{x}(0) = [x_1(0), x_2(0)]^\mathsf{T}$. The solution to this ODE is given by $\mathbf{x}(t) = x_1(0)e^{-t}[1, 0]^\mathsf{T} + x_2(0)e^{-5t}[0, 1]^\mathsf{T}$. For all initial points in $\mathbb{R}^2$, we have $\mathbf{x}(t) \to \mathbf{0}$ as $t \to \infty$. Furthermore, as $t \to \infty$, the trajectory $\mathbf{x}(t)$ for any initial point is approximately parallel to the x-axis. We draw the phase plane in Fig. 1a.*

*Assume that the points on the upper half y-axis belongs to class 1 while we have a linear classifier that seperates class 1 and class 2 as shown in Fig. 1a. We observe that for the initial point A belonging to class 1, the solution from a small perturbed initial point $A + \epsilon$ is misclassified as class 2 for a large enough t for any linear classifier. We see from this example that Lyapunov stability iteself does not imply adversarial robustness in graph neural flow models.*

*This example indicates that Lyapunov stability does not guarantee node classification robustness. Additionally, for a system exhibiting global contraction to a single equilibrium point, structural stability may also be ensured. For instance, in the case of GRAND, even if the edges are perturbed, the system maintains the same number of equilibrium points with global contraction.* We conclude that even an amalgamation of both Lyapunov stability and structural stability may not help the graph's adversarial robustness for node classification.

In the example shown in Fig. 1, we observe that in the case of GRAND when $\alpha > 1$, the node features from different classes tend to become closer to each other as time progresses. This phenomenon can potentially create more vulnerability to adversarial attacks.

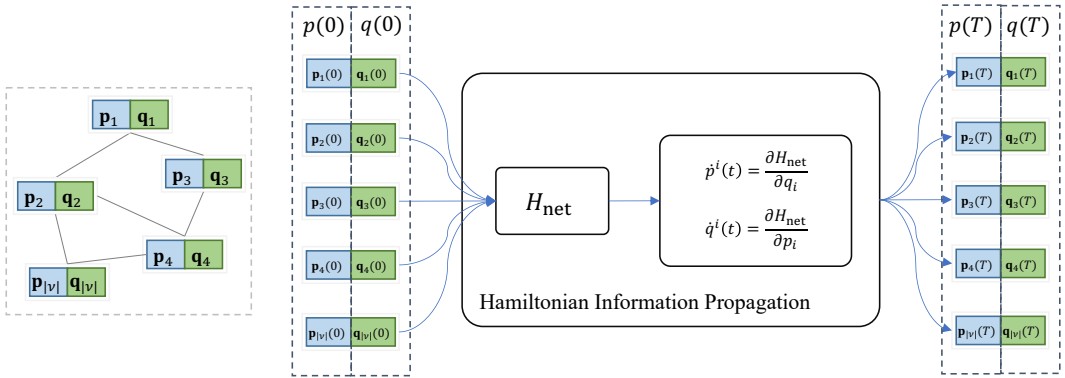

Figure 2: The model architecture: each node is assigned a learnable "momentum" vector at time $t = 0$ which initializes the evolution of the system together with node features. The graph features evolve on following a *learnable* law (10) derived from the $H_{\text{net}}$. At the time $t = T$, we use $q(T)$ as the final node feature. $H_{\text{net}}(q(t), p(t))$ is a learnable graph energy function.

# 4  Hamiltonian-Inspired Graph Neural Flow

Drawing inspiration from the principles of Hamiltonian classical mechanics, we introduce a novel graph neural flow paradigm, namely HamiltoniAN Graph diffusion (HANG). In Hamiltonian mechanics, the notation $(q, p)$ is traditionally used to represent the properties of nodes (position and momentum). Hence, in this section, we adopt this notation instead of $\mathbf{X}$ (as used in Section 3) to denote node features.

## 4.1  Physics Philosophy Behind HANG

In Hamiltonian mechanics, the state evolution of a multi-object physical system, such as an electromagnetic field, double pendulum, or spring network [43, 44], adheres to well-established physical laws. For instance, in a system of charged particles, each particle generates an electromagnetic field that influences other particles. The Hamiltonian for such a system includes terms representing the kinetic and potential energy of each particle, along with terms representing the interactions between particles via their electromagnetic fields. In this paper, we propose a novel concept of information propagation between graph nodes, where interactions follow a similar Hamiltonian style.

In a Hamiltonian system, the position $q$ and momentum $p$ together constitute the phase space $(q, p)$, which comprehensively characterizes the system's evolution. In our HANG model, we process the raw node features of the graph using a linear input layer, yielding $2r$-dimensional vectors. Following the methodologies introduced in the GNN work [31, 35], we split the $2r$ dimensions into two equal halves, with the first half serving as the feature (position) vector and the second half as "momentum" vector that guides the system evolution. Concretely, each node $k$ is associated with an $r$-dimensional feature vector $\mathbf{q}_k(0) = (q_k^1, \ldots, q_k^r)$ and an $r$-dimensional momentum vector $\mathbf{p}_k(0) = (p_k^1, \ldots, p_k^r)$. Subsequently, $\mathbf{q}_k(t)$ and $\mathbf{p}_k(t)$ will evolve along with the propagation of information between graph nodes, with $\mathbf{q}_k(0)$ and $\mathbf{p}_k(0)$ serving as the initial conditions.

Following the modeling conventions in physical systems, we concatenate the feature positions of all $|\mathcal{V}|$ vertices into a single vector, treating it as the system's generalized coordinate within a $r|\mathcal{V}|$-dimensional manifold, a process that involves index relabeling[3].

$$q(t) = \left( q^1(t), \ldots q^{r|\mathcal{V}|}(t) \right) = \left( \mathbf{q}_1(t), \ldots, \mathbf{q}_{|\mathcal{V}|}(t) \right). \qquad (7)$$

This $r|\mathcal{V}|$-dimensional coordinate representation at each time instance provides a snapshot of the state of the graph system. Similarly, we concatenate all the "momentum" vectors at time $t$ to construct an $r|\mathcal{V}|$-dimensional vector:

$$p(t) = \left( p_1(t), \ldots p_{r|\mathcal{V}|}(t) \right) = \left( \mathbf{p}_1(t), \ldots, \mathbf{p}_{|\mathcal{V}|}(t) \right), \qquad (8)$$

---

[3]In multilinear algebra and tensor computation, vector components employ upper indices, while covector components use lower indices. We adhere to this convention.

which can be interpreted as a generalized momentum vector for the entire graph system.

In physics, the system evolves in accordance with fundamental physical laws, and a conserved quantity function $H(q, p)$ remains constant along the system's evolution trajectory. This conserved quantity is typically interpreted as the "system energy". In our HANG model, instead of defining an explicit energy function $H(p, q)$ from a fixed physical law, we utilize a learnable energy function $H_{\mathrm{net}} : \mathcal{G} \to \mathbb{R}^{+}$ parameterized by a neural network, referred to as the *Hamiltonian energy function*:

$$H_{\mathrm{net}} : \mathcal{G} \to \mathbb{R}^{+} \tag{9}$$

We allow the graph features to evolve according to a learnable Hamiltonian law analogous to basic physical laws. More specifically, we model the feature evolution trajectory as the following canonical Hamilton's equations, which is a restatement of (2):

$$\dot{q}(t) = \frac{\partial H_{\mathrm{net}}}{\partial p}, \quad \dot{p}(t) = -\frac{\partial H_{\mathrm{net}}}{\partial q}, \tag{10}$$

with the initial features $(q(0), p(0)) \in \mathbb{R}^{2r|\mathcal{V}|}$ at time $t = 0$ being the vectors after the raw node features transformation.

The neural ODE given by (10) can be trained and solved through integration to obtain the trajectory $(q(t), p(t))$. At the terminal time point $t = T$, the system's solution is represented as $(q(T), p(T))$. We then apply the canonical projection map $\pi$ to extract the nodes' concatenated feature vector $q(T)$ as follows: $\pi((q(T), p(T))) = q(T)$. This concatenated feature vector $q(T)$ is subsequently decompressed into individual node features for utilization in downstream tasks. For this study, we employ backpropagation to minimize the cross-entropy in node classification tasks. The complete model architecture is depicted in Fig. 2, while a comprehensive summary of the full algorithm can be found in the Appendix G.

## 4.2 Hamiltonian Energy Conservation

Referring to [45], it is known that the total Hamiltonian energy $H_{\mathrm{net}}$ remains constant along the trajectory of its induced Hamiltonian flow. This principle is recognized as the law of energy conservation in a Hamiltonian system.

**Theorem 2.** *If the graph system evolves in accordance with* (10)*, the total energy $H_{\mathrm{net}}(q(t), p(t))$ of the system remains constant. BIBO stability is achieved if $H_{\mathrm{net}}$ remains bounded for all bounded inputs and, as $(q, p) \to \infty$, $H_{\mathrm{net}}(q, p) \to \infty$.*

In light of Theorem 2, if our system evolves following (10), it adheres to the law of energy conservation. As a result, our model guarantees conservative stability. Compared to GraphCON, which conserves Dirichlet energy over time $t$ under specific conditions, the notion of Hamiltonian energy conservation is broader in scope. Under the settings delineated in Section 3, GraphCON can be considered as a particular instance of HANG when $H_{\mathrm{net}}$ is set to represent Dirichlet energy $\mathcal{E}(q)$:

**Definition 5** (Dirichlet energy [35])**.** *The Dirichlet energy is defined on node features $q(t)$ at time $t$ of an undirected graph $\mathcal{G}$ as*

$$\mathcal{E}(q(t)) = \frac{1}{|\mathcal{V}|} \sum_{i} \sum_{j \in \mathcal{N}(i)} \|\mathbf{q}_i(t) - \mathbf{q}_j(t)\|^2, \tag{11}$$

*where $\mathcal{N}(i) = \{j : [i, j] \in \mathcal{E}\}$ is the set of neighbors adjacent to node $i$ in the graph.*

## 5 Different Hamiltonian Energy Functions

In physical systems, the system is often depicted as a graph where two neighboring vertices with mass are connected by a spring of given stiffness and length [44]. The system's energy is thus related to the graph's topology. Similarly, in our graph system, the energy function $H_{\mathrm{net}}$ involves interactions between neighboring nodes, signifying the importance of the graph's topology. There exist multiple ways to learn the energy function, and we present two examples below.

Table 1: The stability summary for different graph neural flows where the ✓ denotes that stability is affirmed under additional conditions.

| Graph Neural Flows | BIBO stabiliy | Lyapunov stabiliy | Structural stabiliy | Conservative stabiliy |
|---|---|---|---|---|
| GRAND | ✓ | ✓ | ✓ | ✓ |
| BLEND | ✓ | ✓ | ✓ | ✓ |
| GraphCON | ✓ | ✗ | ✗ | ✓ |
| GraphBel | ✓ | ✓ | ✓ | ✓ |
| HANG | ✗ | ✗ | ✗ | ✓ |
| HANG-quad | ✗ | ✓ | ✗ | ✓ |

## 5.1 Vanilla HANG

We define $H_{\text{net}}$ as a composition of two graph convolutional layers:

$$H_{\text{net}} = \left\| \left( g_{\text{gcn}_2} \circ \tanh \circ g_{\text{gcn}_1} \right) (q, p) \right\|_2 , \tag{12}$$

where $g_{\text{gcn}_1} : \mathbb{R}^{2r \times |\mathcal{V}|} \to \mathbb{R}^{d \times |\mathcal{V}|}$ and $g_{\text{gcn}_2} : \mathbb{R}^{d \times |\mathcal{V}|} \to \mathbb{R}^{|\mathcal{V}|}$ are two GCN [3] layers with different hidden dimensions. A $\tanh$ activation function is applied between the two GCN layers, and $\| \cdot \|_2$ denotes the $\ell_2$ norm. We concatenate $\mathbf{q}_k$ and $\mathbf{p}_k$ for each node $k$ at the input of the above composite function, resulting in an input dimension of $2r$ per node. From Theorem 2, it follows that HANG exhibits BIBO stability. If $(q(t), p(t))$ were unbounded, the value of $H_{\text{net}}$ would also become unbounded, contradicting the energy conservation principle. In subsequent discussions, this invariant is referred to as HANG.

## 5.2 Quadratic HANG (HANG-quad)

The general vanilla HANG with conservative stability does not possess Lyapunov stability. Other additional conditions may be required for Lyapunov stability. The following Lagrange-Dirichlet Theorem provides a sufficient condition.

**Theorem 3** (Lagrange-Dirichlet Theorem [46]). *Let $\mathbf{z}_e$ be a locally quadratic equilibrium of the natural Hamiltonian* (2) *with*

$$H = T(q, p) + U(q), \tag{13}$$

*where $T$ is a positive definite, quadratic function of $p$. Then $\mathbf{z}_e$ is Lyapunov stable if the position of it is a strict local minimum of $U(q)$.*

Theorem 3 implies that we can design an energy function $H_{\text{net}}$ such that the induced graph neural flow is both Lyapunov stable and energy conservative. For instance, we can define $T$ as

$$T(q, p) = \sum_i \mathbf{h}_i^{\mathsf{T}} \mathbf{h}_i + \lambda \mathbf{p}_i^{\mathsf{T}} \mathbf{p}_i, \tag{14}$$

where $\mathbf{h}_i = \sum_{j \in \mathcal{N}(i)} a_G (\mathbf{q}_i, \mathbf{q}_j) \mathbf{p}_j$ is the aggregation of $\mathbf{p}_j$ through the attention mechanism $a_G (\mathbf{q}_i, \mathbf{q}_j)$ calculated based on $q$ (or we directly use adjacency matrix with $a_G (\mathbf{q}_i, \mathbf{q}_j) \equiv \mathbf{W}[i, j]$). The term $\lambda$ is a small positive number included to ensure the positive definiteness. If $U(q)$ has only a single global minimum, such as $U(q) = \|q\|$ with an $\ell_2$ norm, this stability may not ensure adversarial robustness as discussed in Section 3.1. An alternative is setting $U(q) = \|\sin(q)\|$, which fosters Lyapunov stability across multiple local equilibrium points. However, this choice considerably restricts the form that $U$ can take and may consequently limit the model's capacity. In the implementation, we set the function $U(q)$ in $H$ to be a single GAT [6] layer $g_{\text{gat}} : \mathbb{R}^{r \times |\mathcal{V}|} \to \mathbb{R}^{r \times \mathcal{V}}$ with a $\sin$ activation function followed by an $\ell_2$ norm, i.e., $U(q) = \|g_{\text{gat}}(q)\|_2$. The stability of HANG and HANG-quad is summarized in Table 1.

## 6 Experiments

In this section, we conduct a comprehensive evaluation of our theoretical findings and assess the robustness of two conservative stable models: HANG and HANG-Quad. We compare their performance against various benchmark GNN models, including GAT [47], GraphSAGE [48], GCN [3] and other prevalent graph neural flows. We incorporate different types of graph adversarial attacks as described in Section 6.1 and Section 6.3. These attacks are conducted in a black-box setting, where a surrogate model is trained to generate perturbed graphs or features. For more experiments, we direct readers to Appendix C.

Table 2: Node classification accuracy (%) on graph **injection, evasion, non-targeted** attack in **inductive** learning. The best and the second-best result for each criterion are highlighted in red and blue respectively.

| Dataset | Attack | HANG | HANG-quad | GraphCON | GraphBel | GRAND | GAT | GraphSAGE | GCN |
|---|---|---|---|---|---|---|---|---|---|
| Cora | *clean* | 87.13±0.86 | 79.68±0.62 | 86.27±0.51 | 86.13±0.51 | 87.53±0.59 | 87.58±0.64 | 86.65±1.51 | 88.31±0.48 |
| | PGD | 78.37±1.84 | 79.05±0.42 | 42.81±0.30 | 40.04±0.68 | 39.65±1.32 | 38.27±2.73 | 35.43±5.05 | 35.83±0.71 |
| | TDGIA | 79.76±0.99 | 79.54±0.65 | 43.0±0.24 | 39.10±0.80 | 41.77±2.70 | 39.39±7.57 | 34.38±2.25 | 33.05±1.09 |
| | MetaGIA | 77.48±1.02 | 78.28±0.56 | 42.30±0.33 | 39.93±0.59 | 39.36±1.26 | 39.49±4.19 | 38.14±2.23 | 35.84±0.73 |
| Citeseer | *clean* | 74.11±0.62 | 71.85±0.48 | 74.84±0.49 | 69.62±0.56 | 74.98±0.45 | 67.87±4.97 | 63.22±9.14 | 72.63±1.14 |
| | PGD | 72.31±1.16 | 71.07±0.41 | 40.56±0.36 | 55.67±5.35 | 36.68±1.05 | 32.65±3.80 | 32.70±5.11 | 30.69±2.33 |
| | TDGIA | 72.12±0.52 | 71.69±0.40 | 36.67±1.25 | 34.17±4.68 | 36.67±1.25 | 30.53±3.57 | 26.11±2.94 | 21.10±2.35 |
| | MetaGIA | 72.92±0.66 | 71.60±0.48 | 48.36±2.12 | 45.60±4.31 | 46.23±2.01 | 37.68±4.0 | 35.75±5.50 | 35.86±0.68 |
| CoauthorCS | *clean* | 96.16±0.09 | 95.27±0.12 | 95.10±0.12 | 93.93±0.48 | 95.08±0.12 | 92.84±0.41 | 93.0±0.39 | 93.33±0.37 |
| | PGD | 94.80±0.33 | 95.08±0.23 | 42.68±10.13 | 71.35±7.29 | 74.69±12.42 | 34.22±32.58 | 7.29±2.58 | 11.02±5.04 |
| | TDGIA | 95.40±0.13 | 95.09±0.09 | 7.92±4.11 | 43.13±3.58 | 5.05±1.43 | 18.08±15.74 | 6.47±4.25 | 3.61±1.77 |
| | MetaGIA | 94.85±0.31 | 94.83±0.28 | 67.79±4.04 | 73.98±8.26 | 84.31±4.26 | 52.01±24.21 | 7.82±2.69 | 15.70±3.95 |
| Pubmed | *clean* | 89.93±0.27 | 88.10±0.33 | 88.78±0.46 | 86.97±0.37 | 88.44±0.34 | 87.41±1.73 | 88.71±0.37 | 88.46±0.20 |
| | PGD | 81.81±1.94 | 87.69±0.57 | 45.06±1.51 | 46.06±0.97 | 44.61±2.78 | 48.94±12.99 | 44.62±6.49 | 39.03±0.10 |
| | TDGIA | 86.62±1.05 | 87.55±0.60 | 46.30±1.56 | 52.24±0.68 | 44.99±1.10 | 47.56±3.11 | 47.61±0.91 | 42.64±1.41 |
| | MetaGIA | 87.58±0.75 | 87.40±0.62 | 45.55±1.18 | 50.04±0.64 | 44.36±1.20 | 44.75±2.53 | 42.39±0.53 | 40.42±0.17 |

Table 3: Node classification accuracy (%) on graph **injection, evasion, targeted** attack in **inductive** learning.

| Dataset | Attack | HANG | HANG-quad | GraphCON | GraphBel | GRAND | GAT | GraphSAGE | GCN |
|---|---|---|---|---|---|---|---|---|---|
| Computers | PGD | 90.83±0.53 | 87.53±0.99 | 74.01±4.87 | 89.33±0.56 | 65.75±5.00 | 35.72±10.71 | 18.33±0.57 | 17.80±0.06 |
| | TDGIA | 90.88±0.50 | 87.75±0.53 | 80.11±3.21 | 88.21±0.63 | 77.18±7.60 | 66.09±17.07 | 50.89±1.30 | 66.51±3.90 |
| | MetaGIA | 90.73±0.64 | 89.32±0.48 | 87.95±2.65 | 89.27±0.57 | 81.85±4.66 | 60.28±11.11 | 37.51±4.49 | 36.19±0.10 |
| ogbn-arxiv | PGD | 58.32±1.53 | 52.95±0.33 | 53.48±6.60 | OOM | 45.07±3.87 | 52.75±4.55 | 25.48±5.65 | 29.76±2.56 |
| | TDGIA | 66.36±2.50 | 61.93±0.94 | 29.77±8.01 | OOM | 22.69±5.96 | 17.76±13.0 | 4.78±2.76 | 28.43±5.60 |
| | MetaGIA | 64.69±1.77 | 64.83±1.21 | 57.91±4.99 | OOM | 50.75±3.51 | 57.07±4.87 | 33.84±2.51 | 37.36±0.83 |

Table 4: Node classification accuracy (%) under **modification, poisoning, non-targeted** attack (Metattack) in **transductive** learning.

| Dataset | Ptb Rate(%) | HANG | HANG-quad | GraphCON | GraphBel | GRAND | GAT | GCN | RGCN | GCN-SVD | Pro-GNN |
|---|---|---|---|---|---|---|---|---|---|---|---|
| Polblogs | 0 | 94.77±1.07 | 94.63±1.06 | 93.14±0.84 | 85.13±2.22 | 95.57±0.44 | 95.35±0.20 | 95.69±0.38 | 95.22±0.14 | 95.31±0.18 | 93.20±0.64 |
| | 5 | 80.19±2.52 | 94.38±0.82 | 70.24±2.71 | 51.84±3.38 | 77.58±3.46 | 83.69±1.45 | 73.07±0.80 | 74.34±0.19 | 89.09±0.22 | 93.29±0.18 |
| | 10 | 74.92±4.32 | 92.46±1.56 | 71.87±1.71 | 56.54±2.30 | 77.99±1.35 | 76.32±0.85 | 70.72±1.13 | 71.04±0.34 | 81.24±0.49 | 89.42±1.09 |
| | 15 | 71.65±1.34 | 90.85±2.43 | 69.0±0.90 | 53.41±3.08 | 73.84±1.46 | 68.80±1.14 | 64.96±1.91 | 67.28±0.38 | 68.10±3.73 | 86.04±2.21 |
| | 20 | 66.27±3.39 | 89.19±3.72 | 60.46±2.31 | 52.18±0.54 | 69.14±1.32 | 51.50±1.63 | 51.27±1.23 | 59.89±0.34 | 57.33±3.15 | 79.56±5.68 |
| | 25 | 65.80±2.33 | 86.89±8.90 | 58.67±1.87 | 51.39±1.36 | 67.65±1.65 | 51.19±1.49 | 49.23±1.36 | 56.02±0.56 | 48.66±9.93 | 63.18±4.40 |
| Pubmed | 0 | 85.08±0.20 | 85.23±0.14 | 86.65±0.17 | 84.02±0.26 | 85.06±0.26 | 83.73±0.40 | 87.19±0.09 | 86.16±0.18 | 83.44±0.21 | 87.33±0.18 |
| | 5 | 85.08±0.18 | 85.12±0.18 | 86.52±0.14 | 83.91±0.26 | 84.11±0.30 | 78.00±0.44 | 83.09±0.13 | 81.08±0.20 | 83.41±0.15 | 87.25±0.09 |
| | 10 | 85.17±0.23 | 85.05±0.19 | 86.41±0.13 | 84.62±0.26 | 84.24±0.18 | 74.93±0.38 | 81.21±0.09 | 77.51±0.27 | 83.27±0.21 | 87.25±0.09 |
| | 15 | 85.0±0.22 | 85.15±0.17 | 86.21±0.15 | 84.83±0.20 | 83.74±0.34 | 71.13±0.51 | 78.66±0.12 | 73.91±0.25 | 83.10±0.18 | 87.20±0.09 |
| | 20 | 85.20±0.19 | 85.03±0.19 | 86.20±0.18 | 84.89±0.45 | 83.58±0.20 | 68.21±0.96 | 77.35±0.09 | 71.18±0.31 | 83.01±0.22 | 87.09±0.10 |
| | 25 | 85.06±0.17 | 84.99±0.16 | 86.04±0.14 | 85.07±0.15 | 83.66±0.25 | 65.41±0.77 | 75.50±0.17 | 67.95±0.15 | 82.72±0.18 | 86.71±0.09 |

## 6.1 Graph Injection Attacks (GIA)

We implement various GIA algorithms following the methodology in [49]. This framework consists of node injection and feature update procedures. Node injection involves generating new edges for injected nodes using gradient information or heuristics. We use the PGD-GIA method [49] to randomly inject nodes and determine their features with the PGD algorithm [50]. TDGIA [12] identifies topological vulnerabilities to guide edge generation and optimize a smooth loss function for feature generation. MetaGIA [49] performs iterative updates of the adjacency matrix and node features using gradient information. Our datasets include citation networks (Cora, Citeseer, Pubmed) [51], the Coauthor academic network [52], an Amazon co-purchase network (Computers) [52], and the Ogbn-Arxiv dataset [53]. For inductive learning, we follow the data splitting method in the GRB framework [54], with 60% for training, 10% for validation, and 20% for testing. Details on data statistics and attack budgets can be found in Appendix C.1. Targeted attacks are applied to the Ogbn-Arxiv and Computers datasets [49]. Additional results for various attack strengths and white-box attacks can be found in Appendix C.4 and Appendix C.3, respectively.

## 6.2 Performance Results Under GIAs

Upon examining the results in Table 2 and Table 3 pertaining to experiments conducted under GIA conditions, the robustness of our proposed HANG and HANG-quad, is notably prominent across different GIA scenarios. Interestingly, GRAND, despite its Lyapunov stability as analyzed in Theorem 1, does not significantly outperform GAT under certain attacks. In contrast, HANG consistently displays robustness against attacks. Notably, HANG-quad exhibits superior performance to HANG on the Pubmed dataset under GIA perturbations, underscoring the effectiveness of integrating both Lyapunov stability and Hamiltonian mechanics to boost robustness. Although other graph neural flows might demonstrate a range of improved performance compared to conventional GNN models under GIA, the degree of improvement is not consistently distinct. Despite the pronounced association between conservative stability in Hamiltonian systems and adversarial robustness, a clear relationship between adversarial robustness with the other stability of the graph neural flows outlined in Table 1 is not immediately discernible. The performance differential between HANG variants and other graph neural flows further underscores the potential of our proposed Hamiltonian models in enhancing robustness against GIA attacks.

## 6.3 Graph Modification Attacks

To evaluate the robustness of our proposed conservative models, we conducted graph modification adversarial attacks using the Metattack method [17]. We followed the attack setting described in the Pro-GNN [55] and utilized the perturbed graph provided by the library [56] to ensure a fair comparison. The perturbation rate, which indicates the proportion of altered edges, was incrementally varied in 5% increments from 0% to 25%. For comparison, we also considered other defense models for GNNs, namely Pro-GNN [55], RGCN [57], and GCN-SVD [58]. We report the results of baseline models from [55].

## 6.4 Performance Results Under Modification/Poisoning/Transductive Attacks

In the case of the Polblogs dataset [59], as shown in Table 4, our proposed HANG-quad model demonstrates superior performance compared to other methods, including existing defense models. This result indicates that incorporating Lyapunov stability indeed enhances HANG's robustness against graph modification and poisoning attacks. For the Pubmed dataset, we note that the impact of Meta-attacks of varying strengths on *all* graph neural flows, including our proposed ones, is negligible. Conversely, traditional GNN models such as GAT, GCN, and RGCN are marginally affected as the attack strength escalates. This observation underlines the robustness of graph neural flows, including our proposed models, against Meta-attacks on this dataset.

## 6.5 Combination with other defense mechanisms

It merits noting that our models, HANG and HANG-quad, can be readily integrated with additional defense mechanisms against adversarial attacks. These include Adversarial Training (AT) [60] and other preprocessing methods such as GNNGUARD [61]. This integration can further bolster the robustness of the HANG model. To validate this enhancement, extensive experiments are conducted, with results detailed in Appendix C.8 and Appendix C.9.

# 7 Conclusion

In this paper, we conducted a comprehensive study on stability notions in the context of graph neural flows and made significant findings. While Lyapunov stability is frequently employed, it alone may not suffice in guaranteeing robustness against adversarial attacks. With a grounding in foundational physics principles, we proposed a shift towards conservative Hamiltonian neural flows for crafting GNNs resilient against adversarial attacks. Our empirical comparisons across diverse neural flow GNNs, as tested on multiple benchmark datasets subjected to a range of adversarial attacks, have further corroborated this proposition. Notably, GNNs that amalgamate conservative Hamiltonian flows with Lyapunov stability exhibited marked enhancement in their robustness metrics. We are optimistic that our work will inspire further research into marrying physics principles with machine learning paradigms for enhanced security.

# 8 Acknowledgments and Disclosure of Funding

This research is supported by the Singapore Ministry of Education Academic Research Fund Tier 2 grant MOE-T2EP20220-0002, and the National Research Foundation, Singapore and Infocomm Media Development Authority under its Future Communications Research and Development Programme. To improve the readability, parts of this paper have been grammatically revised using ChatGPT [62].

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

# A  Summary of Supplement

In this supplement, we expand on material from the main paper, addressing each point as follows:

1. An in-depth comparison with related work is covered in Appendix B.

2. Appendix C presents the datasets and details of the attackers, and includes additional experimental results, inference time, and model size to reinforce our model's effectiveness.

3. Theoretical proofs supporting assertions made in the main paper are presented in Appendix E.

4. Further discussions on the robustness insights of our model are covered in Appendix D.

5. The complete summary of the algorithm can be found in Appendix G.

6. Lastly, we address the limitations of our work and discuss its broader impact.

# B  Related Work

In what follows, we briefly review a few concepts closely related to our work.

**Graph Adversarial Attacks and Defenses.** In *modification attacks*, adversaries can perturb a graph's topology by adding or removing edges [9, 14–17, 63–66]. To improve the modification attack performance, adversaries are also permitted to perturb node attributes [9, 17, 63, 64, 67, 68]. In *injection attacks*, adversaries can only inject malicious nodes into the original graph [10–13] while the edges and nodes inside the original graph are not allowed to be perturbed. For *defense methods* against adversarial attacks, multiple robust GNN models have been proposed. Examples include RobustGCN [69], GRAND [70], ProGNN [71], GLNN [72], GAUGM [73], STABLE [74] and RWL-GNN [75]. In addition, preprocessing-based defenders including GNN-SVD [58] and GNNGuard [76] may help to improve GNN robustness.

*In this paper, we use different attacks to test GNNs robustness. We compare graph neural flows with different stability settings with the above-mentioned defense methods as robustness benchmarks.*

**Stable Graph Neural Flow Networks.** While traditional GNNs perform message passing on a simple, discrete and flat space, graph neural flows model the message passing as a continuous diffusion process that occurs on a smooth manifold. GRAND [29] and GRAND++ [30] use heat diffusion to achieve feature information exchange. BLEND [31] exploits the Beltrami diffusion where the nodes' positional information is updated along with their features. GraphCON [35] adopts the coupled oscillator model that preserves the graph's Dirichlet energy over time and thus mitigates the oversmoothing problem. In general, [32] shows that graph neural PDEs are Lyapunov stable and exhibit stronger robustness against graph topology perturbation than traditional GNNs.

*While most of the above-mentioned graph neural flows are Lyapunov stable, whether the notion of Lyapunov stability leads to better adversarial robustness is an open question. In this paper, we argue that Lyapunov stability does not necessarily imply adversarial robustness.*

**Hamiltonian Neural Networks.** Hamiltonian equations have been applied to conserve an energy-like quantity in (graph) neural networks. The references [26, 27, 77] train a neural network to infer the Hamiltonian dynamics of a physical system, where Hamiltonian equations are solved using neural ODE solvers. In [78], the authors propose to learn a Hamiltonian function of the system by a neural network to capture the dynamics of physical systems from observed trajectories. They shows that the network performs well on noisy and complex systems such as a spring-chain system. To forecast dynamics, the work [79] use neural networks that incorporate Hamiltonian dynamics to efficiently learn phase space orbits and demonstrate the effectiveness of Hamiltonian neural networks on several dynamics benchmarks. The paper [80] builds a Hamiltonian-inspired neural ODE to stabilize the gradients so as to avoid gradient vanishing and gradient exploding.

*In this paper, inspired by existing Hamiltonian neural networks, we introduce several energy-conservative graph neural flows. We are neither simulating a physical system nor forecasting a forecast the dynamics for a physical problem. Instead, we combine the Hamiltonian mechanics concept with graph neural networks to develop a new robust GNN.*

# C More Experiments

## C.1 Data and Attackers

The datasets and attack budgets utilized in Table 2 and Table 3 are outlined in Table 5 and Table 6, respectively. These datasets span various domains and scales, thereby providing a diverse base for our study. The adopted attack budget aligns consistently with the specifications set out in the paper [49].

Table 5: Dataset Details

| Dataset | # Nodes | # Edges | # Features | # Classes |
|---------|---------|---------|------------|-----------|
| Cora | 2708 | 5429 | 1433 | 7 |
| Citeseer | 3327 | 4732 | 3703 | 6 |
| PubMed | 19717 | 44338 | 500 | 3 |
| Coauthor | 18,333 | 81,894 | 6,805 | 15 |
| Computers | 13,752 | 245,861 | 767 | 10 |
| Ogbn-Arxiv | 169343 | 1166243 | 128 | 40 |

Table 6: Attacks' budgets for GIA. $^*$ refers to targted GIA.

| Dataset | max # Nodes | max # Edges |
|---------|-------------|-------------|
| Cora | 60 | 20 |
| Citeseer | 90 | 10 |
| PubMed | 200 | 100 |
| Coauthor | 300 | 150 |
| Ogbn-Arxiv$^*$ | 120 | 100 |
| Computers$^*$ | 100 | 150 |

## C.2 Implementation Details

The raw node features are compressed to a fixed dimension, such as 64, using a fully connected (FC) layer to generate the initial features $q(0)$ in (10). At time $t = 0$, $p(0)$ and $q(0)$ are initialized identically. For $t > 0$, both $q(t)$ and $p(t)$ undergo updates using a graph ODE. The ODE is solved using the solver from [18]. It is observed that different solvers deliver comparable performance in terms of clean accuracy. However, to mitigate computational expense, the Euler solver is employed in our experiments, with an ablation study on different solvers provided for further insight. The integral time $T$ acts as a hyperparameter in our model. Interestingly, the performance of the model exhibits minimal sensitivity to this time $T$. For all datasets, we establish the time $T$ as 3 and maintain a fixed step size of 1. This setup aligns with a fair comparison to three-layer GNNs.

All the baseline models presented in Table 2 and Table 3 are implemented based on the original work of [49]. The baseline model results in Table 4 are directly extracted from the paper [55]. This is done as we employ the same clean and perturbed graph datasets provided in their research [55].

Our experiment code is developed based on the following repositories:

- `https://github.com/tk-rusch/GraphCON`
- `https://github.com/twitter-research/graph-neural-pde`
- `https://github.com/LFhase/GIA-HAO`
- `https://github.com/ChandlerBang/Pro-GNN`

## C.3 White-box Attack

In our main study, we utilized black-box attacks. Now, we extend our experiments to incorporate white-box, injection, and evasion attacks. In the context of white-box attacks, the adversaries have full access to the target model, enabling them to directly attack the target model to generate a perturbed graph. This represents a significantly more potent form of attack than the black-box variant. Moreover, we execute inductive learning tasks the same as Table 2, with the corresponding results reported in Table 7. We observe that, under white-box attack conditions, all other baseline models exhibit severely reduced performance, essentially collapsing across all datasets. The classification accuracy of the HANG model experiences a slight decline on the Cora, Citeseer, and Pubmed datasets. However, its performance remains substantially superior to other graph neural flow models or GNN models. Intriguingly, our HANG-quad model remains virtually unaffected by the white-box attacks, maintaining a performance level similar to that observed under black-box attacks. Such robustness of HANG-quad underscores the pivotal contribution of Lyapunov stability when combined with our Hamiltonian-driven design.

Table 7: Node classification accuracy (%) on graph **injection, evasion, non-targeted, white-box** attack in **inductive** learning. The best and the second-best result for each criterion are highlighted in red and blue respectively.

| Dataset | Attack | HANG | HANG-quad | GraphCON | GraphBel | GRAND | GAT | GraphSAGE | GCN |
|---|---|---|---|---|---|---|---|---|---|
| Cora | clean | 87.13±0.86 | 79.68±0.62 | 86.27±0.51 | 86.13±0.51 | 87.53±0.59 | 87.58±0.64 | 86.65±1.51 | 88.31±0.48 |
| | PGD | 67.69±3.84 | 78.04±0.91 | 42.09±1.74 | 37.16±1.69 | 36.02±4.09 | 30.95±8.22 | 29.79±7.56 | 35.83±0.71 |
| | TDGIA | 64.54±3.95 | 77.35±0.66 | 19.01±1.45 | 15.46±1.98 | 14.72±1.97 | 4.81±1.19 | 17.83±6.62 | 33.05±1.09 |
| Citeseer | clean | 74.11±0.62 | 71.85±0.48 | 74.84±0.49 | 69.62±0.56 | 74.98±0.45 | 67.87±4.97 | 63.22±9.14 | 72.63±1.14 |
| | PGD | 67.54±1.52 | 72.21±0.71 | 42.78±1.54 | 32.24±1.21 | 38.57±1.94 | 25.87±6.69 | 29.65±4.11 | 30.69±2.33 |
| | TDGIA | 63.29±3.15 | 70.62±0.96 | 33.55±1.10 | 16.26±1.20 | 30.11±1.43 | 17.46±3.34 | 17.83±1.56 | 21.10±2.35 |
| CoauthorCS | clean | 96.16±0.09 | 95.27±0.12 | 95.10±0.12 | 93.93±0.48 | 95.08±0.12 | 92.84±0.41 | 93.0±0.39 | 93.33±0.37 |
| | PGD | 93.40±0.71 | 93.25±1.02 | 7.80±1.18 | 13.21±4.21 | 8.0±0.06 | 11.96±7.10 | 10.73±6.84 | 11.02±5.04 |
| | TDGIA | 93.38±0.71 | 94.12±0.43 | 7.35±1.61 | 10.38±1.07 | 4.53±1.33 | 1.35±0.55 | 2.89±1.65 | 3.61±1.77 |
| Pubmed | clean | 89.93±0.27 | 88.10±0.33 | 88.78±0.46 | 86.97±0.37 | 88.44±0.34 | 87.41±1.73 | 88.71±0.37 | 88.46±0.20 |
| | PGD | 68.62±2.82 | 87.64±0.39 | 36.86±2.63 | 39.34±0.77 | 39.52±3.35 | 38.04±4.91 | 38.76±4.58 | 39.03±0.10 |
| | TDGIA | 69.56±3.16 | 87.91±0.46 | 31.49±1.87 | 30.15±1.30 | 36.19±7.04 | 24.43±4.10 | 38.89±0.76 | 42.64±1.41 |

## C.4 Attack Strength

We assess the robustness of the HANG model and its variant under varying attack strengths, with the node classification results displayed in Table 8. Our analysis reveals that the HANG model demonstrates superior robustness as the attack budget escalates. It is noteworthy, however, that under larger attack budgets, HANG-quad may relinquish its robustness attribute in the face of PGD and Meta injection attacks. This result implies that the amalgamation of Lyapunov stability and conservative Hamiltonian system facilitates enhanced robustness only under minor graph perturbations. On the contrary, the HANG model, which solely incorporates the conservative Hamiltonian design, exhibits a consistently high-performance level regardless of the increasing attack strength.

Table 8: Node classification accuracy (%) on graph **injection, evasion, non-targeted, black-box** attack in **inductive** learning under various attack strength.

| Dataset | Attack | # nods/edges injected | HANG | HANG-quad | GraphCON | GraphBel | GRAND | GAT | GraphSAGE | GCN |
|---|---|---|---|---|---|---|---|---|---|---|
| Cora | PGD | 80/40 | 75.78±3.46 | 78.41±0.71 | 33.60±2.62 | 33.74±2.84 | 34.04±1.87 | 32.83±3.51 | 26.79±6.53 | 31.31±0.06 |
| | PGD | 100/60 | 74.25±1.39 | 76.74±0.48 | 62.69±0.83 | 33.65±2.50 | 32.72±1.47 | 31.25±2.13 | 25.94±7.24 | 31.23±0.0 |
| | PGD | 120/80 | 71.01±2.60 | 70.69±1.83 | 33.46±1.95 | 34.88±3.66 | 32.21±0.83 | 31.28±0.15 | 25.57±7.28 | 31.23±0.0 |
| | PGD | 140/100 | 69.72±2.94 | 59.75±3.57 | 34.02±1.97 | 36.98±4.87 | 33.31±1.79 | 30.26±3.49 | 25.11±7.56 | 31.23±0.0 |
| | PGD | 160/120 | 68.12±2.93 | 45.06±4.77 | 33.34±2.27 | 39.78±4.15 | 33.07±1.34 | 31.28±0.15 | 24.87±7.80 | 31.23±0.0 |
| | PGD | 180/140 | 66.51±4.17 | 36.69±4.02 | 33.19±1.74 | 45.11±4.76 | 33.12±1.64 | 31.41±0.52 | 24.79±7.90 | 31.23±0.0 |
| | PGD | 200/160 | 66.01±4.10 | 29.58±5.11 | 32.99±2.40 | 50.51±3.94 | 32.18±0.89 | 31.47±0.70 | 24.74±7.96 | 31.23±0.0 |
| | TDGIA | 80/40 | 79.47±2.57 | 78.48±0.54 | 21.69±1.51 | 28.70±3.18 | 22.34±2.18 | 25.33±6.27 | 25.43±3.22 | 26.51±3.36 |
| | TDGIA | 100/60 | 78.73±1.20 | 78.74±0.94 | 24.32±4.16 | 36.07±2.27 | 27.10±2.04 | 27.67±5.38 | 30.63±0.62 | 30.68±0.34 |
| | TDGIA | 120/80 | 78.48±1.91 | 78.20±0.78 | 29.22±1.91 | 36.43±2.58 | 29.06±2.24 | 29.67±6.34 | 30.37±0.41 | 31.28±0.87 |
| | TDGIA | 140/100 | 79.40±2.20 | 77.42±0.71 | 23.54±1.49 | 25.54±4.26 | 27.04±3.25 | 26.99±6.05 | 28.85±2.75 | 31.94±3.26 |
| | TDGIA | 160/120 | 79.37±1.30 | 78.05±1.23 | 24.27±3.00 | 32.40±2.37 | 24.56±2.92 | 19.74±6.83 | 29.78±1.66 | 30.09±0.88 |
| | TDGIA | 180/140 | 78.49±2.34 | 76.90±0.57 | 31.20±1.34 | 41.67±6.30 | 31.44±0.32 | 30.15±4.67 | 31.23±0.0 | 31.23±0.0 |
| | TDGIA | 200/160 | 78.85±2.22 | 76.70±1.02 | 24.94±3.18 | 34.32±1.11 | 29.37±1.49 | 24.64±6.93 | 31.07±0.34 | 31.14±0.38 |
| | MetaGIA | 80/40 | 74.41±2.74 | 78.10±0.71 | 34.73±3.45 | 33.58±2.74 | 33.43±1.20 | 32.16±1.71 | 29.58±3.90 | 31.28±0.10 |
| | MetaGIA | 100/60 | 73.23±2.64 | 76.01±0.64 | 33.90±3.43 | 33.48±3.02 | 33.0±1.53 | 30.30±3.02 | 28.48±3.78 | 31.22±0.04 |
| | MetaGIA | 120/80 | 73.63±1.86 | 18.21±7.41 | 32.91±1.47 | 47.80±4.06 | 32.79±2.59 | 31.23±0.0 | 24.76±7.93 | 31.23±0.0 |
| | MetaGIA | 140/100 | 72.05±2.87 | 59.47±2.71 | 33.11±1.97 | 38.77±5.09 | 33.12±1.05 | 31.65±1.22 | 26.46±5.89 | 31.23±0.0 |
| | MetaGIA | 160/120 | 70.93±2.20 | 17.78±7.89 | 33.51±2.36 | 57.77±3.18 | 32.46±1.0 | 31.42±0.60 | 24.69±8.02 | 31.23±0.0 |
| | MetaGIA | 180/140 | 67.37±4.45 | 30.31±0.33 | 32.42±1.01 | 47.48±4.76 | 33.07±0.94 | 29.15±6.26 | 25.42±7.15 | 31.23±0.0 |
| | MetaGIA | 200/160 | 68.44±2.71 | 16.69±6.29 | 32.68±1.38 | 64.38±1.73 | 32.52±1.04 | 28.79±7.09 | 24.69±8.02 | 31.23±0.0 |
| Citeseer | PGD | 110/30 | 72.23±0.79 | 71.98±0.70 | 25.06±3.41 | 41.09±14.36 | 27.48±3.54 | 18.49±1.94 | 20.55±4.44 | 18.63±0.92 |
| | PGD | 130/50 | 71.61±0.84 | 71.44±0.72 | 22.90±4.73 | 43.05±13.60 | 26.42±6.16 | 18.33±1.75 | 19.16±2.95 | 17.85±1.66 |
| | PGD | 150/70 | 72.01±0.68 | 70.33±0.97 | 24.28±4.68 | 38.89±13.86 | 30.63±4.89 | 18.15±1.77 | 18.73±2.19 | 17.42±0.51 |
| | PGD | 170/90 | 71.22±0.67 | 67.81±1.11 | 23.79±4.70 | 26.14±4.72 | 21.49±2.54 | 19.09±2.83 | 17.51±0.41 | 17.65±0.75 |
| | PGD | 190/110 | 71.18±0.54 | 62.26±1.51 | 22.88±3.01 | 29.57±7.91 | 19.95±2.03 | 17.31±3.40 | 17.44±0.34 | 17.85±1.15 |
| | PGD | 210/130 | 71.13±0.72 | 50.15±1.88 | 24.90±3.30 | 31.26±4.51 | 20.65±1.23 | 17.33±3.40 | 17.39±0.33 | 17.67±0.90 |
| | PGD | 230/150 | 71.13±0.85 | 36.03±2.13 | 28.16±3.98 | 35.71±4.84 | 21.36±2.23 | 17.31±3.40 | 17.38±0.34 | 17.75±1.28 |
| | TDGIA | 110/30 | 72.48±0.67 | 72.30±0.82 | 24.30±1.73 | 26.31±1.64 | 23.85±1.25 | 19.26±3.59 | 20.24±1.97 | 18.80±2.45 |
| | TDGIA | 130/50 | 72.26±0.72 | 72.32±0.69 | 27.34±2.72 | 32.0±2.62 | 23.37±1.07 | 18.15±1.68 | 21.16±2.68 | 20.08±2.78 |
| | TDGIA | 150/70 | 72.15±0.58 | 72.41±0.86 | 26.41±1.68 | 27.39±1.25 | 22.21±1.82 | 19.26±3.08 | 19.54±2.25 | 20.39±2.18 |
| | TDGIA | 170/90 | 72.41±0.64 | 72.06±0.98 | 21.63±1.18 | 25.25±1.94 | 20.05±1.46 | 18.16±2.49 | 19.09±2.28 | 19.37±2.30 |
| | TDGIA | 190/110 | 72.80±0.85 | 72.39±0.65 | 24.56±2.34 | 26.97±2.04 | 19.94±1.44 | 18.04±1.59 | 20.13±1.72 | 21.53±2.92 |
| | TDGIA | 210/130 | 72.35±0.42 | 71.78±0.82 | 24.58±3.51 | 24.49±1.19 | 21.46±1.93 | 18.49±2.13 | 20.16±2.43 | 18.46±1.72 |
| | TDGIA | 230/150 | 71.77±0.71 | 71.76±0.81 | 19.01±2.74 | 28.85±1.65 | 19.28±1.72 | 18.97±2.65 | 20.76±4.0 | 18.76±2.02 |
| | MetaGIA | 110/30 | 72.22±1.38 | 72.43±0.67 | 22.48±2.28 | 20.95±2.29 | 30.29±4.58 | 21.77±2.91 | 22.18±2.88 | 18.72±0.27 |
| | MetaGIA | 130/50 | 72.18±1.17 | 71.96±0.86 | 23.28±4.32 | 19.81±1.44 | 25.91±3.65 | 20.44±2.63 | 20.67±2.39 | 18.16±0.16 |
| | MetaGIA | 150/70 | 71.83±0.98 | 71.05±0.93 | 24.68±4.59 | 19.72±1.82 | 26.82±2.76 | 19.64±2.85 | 20.24±3.80 | 18.19±0.07 |
| | MetaGIA | 170/90 | 71.76±1.22 | 68.81±0.77 | 22.48±2.22 | 20.80±1.97 | 29.07±8.81 | 18.82±2.68 | 19.20±2.75 | 18.27±0.0 |
| | MetaGIA | 190/110 | 71.85±0.60 | 64.54±0.78 | 23.29±4.28 | 22.91±2.98 | 26.28±2.84 | 18.59±2.01 | 18.70±2.24 | 18.28±0.03 |
| | MetaGIA | 210/130 | 71.39±0.72 | 56.44±2.42 | 22.56±3.09 | 24.97±2.53 | 26.84±3.41 | 18.71±2.64 | 18.61±2.09 | 18.27±0.0 |
| | MetaGIA | 230/150 | 71.52±0.65 | 13.48±3.49 | 24.13±3.08 | 48.37±2.85 | 26.81±3.78 | 18.58±1.83 | 18.59±2.06 | 18.27±0.0 |

## C.5 Nettack

We further evaluate the robustness of our model under the targeted poisoning attack, Nettack [9]. Adhering to the settings outlined in [55], we select nodes in the test set with a degree greater than 10 to be the target nodes. We then vary the number of perturbations applied to each targeted node from 1 to 5, incrementing in steps of 1. It is important to note that Nettack only involves feature perturbations. The test accuracy in Table 9 refer to the classification accuracy on the targeted nodes. As demonstrated in Table 9, our HANG-quad model exhibits exceptional resistance to Nettack, thereby underlining its superior robustness. This suggests that the combination of Lyapunov stability and conservative Hamiltonian design significantly enhances robustness in the face of graph poisoning attacks. On its own, Lyapunov stability has been recognized to defend against only slight feature perturbations in the input graph [32]. Furthermore, our HANG model also displays superior resilience compared to other graph neural flows, reinforcing the fact that the Hamiltonian principle contributes to its robustness.

Table 9: Node classification accuracy (%) under **modification, poisoning** targeted attack (Nettack) in **transductive** learning. The best and the second-best result for each criterion are highlighted in red and blue respectively.

| Dataset | Ptb | HANG | HANG-quad | GraphCON | GraphBel | GRAND | GAT | GCN | RGCN | GCN-SVD | Pro-GNN |
|---|---|---|---|---|---|---|---|---|---|---|---|
| Cora | 1 | 75.54±3.10 | 76.99±3.16 | 73.25±3.91 | 63.73±2.25 | 80.12±1.81 | 76.04±2.08 | 75.06±1.02 | 76.75±1.71 | 77.23±1.82 | 81.81±1.66 |
| | 2 | 73.73±3.64 | 76.51±2.60 | 67.83±3.0 | 62.41±2.94 | 76.27±1.79 | 70.24±1.43 | 70.60±1.10 | 70.96±1.14 | 72.53±1.60 | 75.90±1.43 |
| | 3 | 68.43±4.23 | 73.13±2.85 | 68.19±2.10 | 61.20±3.08 | 70.48±3.74 | 65.54±1.34 | 67.95±1.72 | 66.51±1.60 | 66.75±1.54 | 70.12±1.93 |
| | 4 | 66.02±2.21 | 72.53±2.14 | 57.59±2.34 | 56.51±2.72 | 65.30±2.40 | 61.69±0.90 | 61.57±1.47 | 59.28±2.68 | 60.72±1.63 | 65.66±1.35 |
| | 5 | 60.12±3.63 | 68.80±2.55 | 55.30±4.77 | 51.93±2.77 | 57.95±2.38 | 58.31±2.03 | 55.54±1.66 | 55.30±1.66 | 57.71±1.82 | 64.34±1.72 |
| Citeseer | 1 | 76.03±3.51 | 79.05±1.38 | 76.03±3.44 | 68.89±2.67 | 80.0±1.05 | 81.27±1.38 | 78.41±1.62 | 78.25±0.73 | 80.16±2.04 | 81.75±0.79 |
| | 2 | 74.76±2.50 | 77.94±2.29 | 68.73±6.62 | 67.62±3.11 | 74.28±7.47 | 77.43±4.89 | 74.92±3.54 | 75.40±2.04 | 79.84±0.73 | 81.27±0.95 |
| | 3 | 74.76±2.18 | 77.14±2.48 | 60.47±5.24 | 60.63±3.87 | 57.14±9.28 | 60.85±2.99 | 63.97±3.69 | 60.31±1.19 | 77.14±2.86 | 79.68±1.98 |
| | 4 | 73.49±3.02 | 78.41±1.62 | 55.55±6.23 | 53.17±6.48 | 59.84±2.75 | 61.59±4.64 | 55.40±2.60 | 55.49±1.75 | 69.52±3.31 | 77.78±2.84 |
| | 5 | 72.06±3.56 | 73.49±3.48 | 51.75±2.77 | 48.73±4.60 | 48.41±8.10 | 55.56±6.28 | 47.62±5.17 | 47.44±2.01 | 69.21±2.48 | 71.27±4.99 |
| Polblogs | 1 | 97.06±0.66 | 97.37±0.37 | 87.07±1.35 | 68.17±3.25 | 96.41±0.87 | 97.22±0.25 | 96.83±0.17 | 97.00±0.07 | 97.56±0.20 | 96.83±0.06 |
| | 2 | 96.39±1.16 | 96.89±0.16 | 82.92±1.53 | 65.48±2.85 | 92.93±4.21 | 96.11±0.65 | 95.61±0.20 | 95.87±0.23 | 97.12±0.09 | 97.17±0.12 |
| | 3 | 96.02±0.93 | 96.65±0.15 | 80.76±0.74 | 62.59±1.99 | 91.96±4.22 | 95.81±0.56 | 95.41±0.18 | 95.59±0.27 | 96.61±0.14 | 96.93±0.12 |
| | 4 | 93.81±3.86 | 96.26±0.53 | 77.46±1.59 | 58.68±0.40 | 86.83±6.28 | 94.80±0.66 | 94.24±0.24 | 94.37±0.26 | 96.17±0.19 | 96.89±0.16 |
| | 5 | 91.65±4.67 | 95.91±0.33 | 75.30±2.71 | 59.02±3.19 | 83.69±5.88 | 93.28±1.43 | 93.00±0.48 | 93.20±0.43 | 95.13±0.25 | 96.13±0.25 |

## C.6 ODE solvers

The results from various ODE solvers are depicted in Table 10. We consider fixed-step Euler and RK4, along with adaptive-step Dopri5, from [18], and Symplectic-Euler from [35]. The Symplectic-Euler method, being inherently energy-conserving, is particularly suited for preserving the dynamic properties of Hamiltonian systems over long times. Our observations suggest that while the choice of solver slightly influences the clean accuracy for some models, their performance under attack conditions remains fairly consistent. Consequently, there was no specific optimization for solver selection during our experiments. For computational efficiency, we opted for the Euler ODE solver in all experiments presented in the main paper.

Table 10: Node classification accuracy (%) on graph **injection, evasion, non-targeted, black-box** attack in **inductive** learning of Citeseer dataset.

| Attack | Solver | HANG | HANG-quad | GraphCON | GraphBel | GRAND |
|---|---|---|---|---|---|---|
| clean | Euler | 74.11±0.62 | 71.85±0.48 | 74.84±0.49 | 69.62±0.56 | 74.98±0.45 |
| | rk4 | 73.71±1.58 | 72.63±0.56 | 75.80±0.38 | 74.46±0.68 | 75.32±0.78 |
| | symplectic euler | 71.35±1.91 | 72.80±0.64 | 75.43±0.62 | 69.87±0.78 | 75.70±0.71 |
| | dopri5 | 75.20±0.93 | – | 76.24±0.76 | 74.63±0.70 | 75.14±0.56 |
| PGD | Euler | 72.31±1.16 | 71.07±0.41 | 40.56±0.36 | 55.67±5.35 | 36.68±1.05 |
| | rk4 | 71.85±2.04 | 72.38±0.52 | 41.26±0.89 | 41.51±1.76 | 41.21±1.57 |
| | symplectic euler | 70.57±1.74 | 72.46±0.53 | 40.87±2.62 | 49.09±7.82 | 39.72±1.54 |
| | dopri5 | 73.59±0.38 | – | 42.20±2.21 | 40.07±0.87 | 40.53±1.14 |
| TDGIA | Euler | 72.12±0.52 | 71.69±0.40 | 36.67±1.25 | 34.17±4.68 | 36.67±1.25 |
| | rk4 | 71.03±1.64 | 72.85±0.78 | 36.19±2.03 | 37.90±1.70 | 34.21±1.63 |
| | symplectic euler | 71.79±2.03 | 73.31±0.58 | 35.32±1.78 | 28.40±0.91 | 35.12±1.62 |
| | dopri5 | 72.14±0.71 | – | 38.04±1.71 | 40.63±1.60 | 34.39±1.19 |
| MetaGIA | Euler | 72.92±0.66 | 71.60±0.48 | 48.36±2.12 | 45.60±4.31 | 46.23±2.01 |
| | rk4 | 70.25±1.45 | 72.39±0.61 | 42.57±1.09 | 43.38±0.88 | 41.01±0.92 |
| | symplectic euler | 71.56±1.07 | 72.86±0.72 | 42.57±1.09 | 44.72±6.28 | 41.0±0.64 |
| | dopri5 | 71.83±1.26 | – | 42.61±0.57 | 42.93±0.61 | 41.74±0.69 |

## C.7 Computation Time

The average inference time and model size for different models used in our study are outlined in Table 11. This analysis is performed using the Cora dataset, with all graph PDE models employing the Euler Solver, an integration time of 3, and a step size of 1. Additionally, for fair comparison, all baseline models are configured with 3 layers. Upon examination, it is observed that our HANG and HANG-quad models necessitate more inference time compared to other baseline models. This is primarily due to the requirement in these models to initially calculate the derivative. However, when compared to other defense models, such as GCNGUAD, our models are still more efficient, thus validating their practical utility.

Table 11: Average inference time and model size

| Model | HANG | HANG-quad | GraphCON | GraphBel | GRAND | GAT | GraphSAGE | GCN | GCNGUARD | RGCN |
|---|---|---|---|---|---|---|---|---|---|---|
| Inference Time (ms) | 17.16 | 14.41 | 2.60 | 6.18 | 2.25 | 4.13 | 3.85 | 3.93 | 95.90 | 2.50 |
| Model Size(MB) | 0.895 | 0.936 | 0.732 | 0.734 | 0.732 | 0.381 | 0.380 | 0.380 | 0.380 | 0.735 |

## C.8 Combination with Adversarial Training(AT)

Adversarial training (AT) [60] is a technique employed to improve the robustness of machine learning models by exposing them to adversarial examples during the training phase. These adversarial examples are maliciously crafted inputs designed to mislead the model, thereby helping it to learn more robust and generalized features.

In this context, we conducted additional experiments applying PGD-AT [60] to both GAT and HANG models, evaluating their robust accuracy under PGD, TDGIA, and METAGIA attacks in both white-box and black-box scenarios. As illustrated in Table 12 and Table 13, while PGD-AT enhances the robustness of all models, its effect on GAT's resilience to the unique white-box METAGIA attack is notably restrained, highlighting the limitations of PGD-AT's efficacy, particularly when faced with non-AT tactics. Conversely, HANG consistently displays remarkable performance across a variety of attacks, showing significant improvement when combined with PGD-AT. Both HANG-AT and HANG-quad-AT exhibit substantial resistance to these attacks, underscoring HANG's inherent robustness.

Furthermore, we explored adversarial training from a graph spectral perspective, incorporating Spectral Adversarial Training (SAT) [81] into our HANG framework, thus creating the HANG-SAT variant. The integration of SAT, as depicted in Table 12 and Table 13, significantly boosts HANG's robustness against various attacks, with minimal reduction in post-attack accuracy, further exemplifying HANG's intrinsic resilience.

Table 12: Node classification accuracy (%) on graph **injection, evasion, non-targeted, black-box** attack in **inductive** learning.

| Dataset | Attack | HANG | HANG-AT | GAT | GAT-AT | HANG-SAT |
|---|---|---|---|---|---|---|
| Pubmed | *clean* | 89.93±0.27 | 89.45±0.33 | 87.41±1.73 | 85.88±0.47 | 88.38±0.48 |
|  | PGD | 81.81±1.94 | 89.03±0.19 | 48.94±12.99 | 77.78±8.39 | 87.65±1.77 |
|  | TDGIA | 86.62±1.05 | 87.16±3.89 | 47.56±3.11 | 79.56±5.76 | 87.55±2.11 |
|  | MetaGIA | 87.58±0.75 | 88.93±0.18 | 44.75±2.53 | 81.86±3.44 | 88.22±0.22 |

Table 13: Node classification accuracy (%) on graph **injection, evasion, non-targeted, white-box** attack in **inductive** learning.

| Dataset | Attack | HANG | HANG-AT | GAT | GAT-AT | HANG-SAT |
|---|---|---|---|---|---|---|
| Pubmed | *clean* | 89.93±0.27 | 89.42±0.22 | 87.41±1.73 | 85.90±0.44 | 88.38±0.48 |
|  | PGD | 68.62±2.82 | 88.86±0.15 | 38.04±4.91 | 80.85±4.51 | 88.19±0.28 |
|  | TDGIA | 69.56±3.16 | 88.98±0.20 | 24.43±4.10 | 82.16±4.10 | 88.28±0.34 |
|  | MetaGIA | 84.64±1.20 | 88.61±0.20 | 40.02±1.34 | 44.37±6.48 | 85.76±4.12 |

## C.9 Combination with GNN defense mechanisms

We acknowledge the potential advantages of integrating defense mechanisms that operate through diverse strategies. In this context, we combine HANG with GNNGuard [61], a preprocessing-based defense renowned for its effectiveness in enhancing GNN robustness, and GARNET [82], another noteworthy defense mechanism.

We conduct experiments to assess the performance of HANG when amalgamated with GNNGuard and GARNET. The results in Table 14 and Table 15 clearly indicate that such integration effectively enhances the model's robustness. This outcome highlights the adaptability of our methodology with current defense strategies and sheds light on the potential for cooperative improvements in model defense.

Table 14: Node classification accuracy (%) on graph **injection, evasion, non-targeted** attack in **inductive** learning.

| Dataset | Attack | HANG | HANG-GUARD | HANG-quad | HANG-quad-GUARD |
|---|---|---|---|---|---|
| Cora | *clean* | 87.13±0.86 | 86.54±0.57 | 79.68±0.62 | 81.23±0.70 |
| | PGD | 78.37±1.84 | 86.23±0.55 | 79.05±0.42 | 80.91±0.67 |
| | TDGIA | 79.76±0.99 | 85.56±0.34 | 79.54±0.65 | 81.11±0.76 |
| | MetaGIA | 77.48±1.02 | 86.0±0.60 | 78.28±0.56 | 80.10±0.53 |
| Citeseer | *clean* | 74.11±0.62 | 75.95±0.66 | 71.85±0.48 | 73.15±0.61 |
| | PGD | 72.31±1.16 | 75.38±0.82 | 71.07±0.41 | 73.07±0.63 |
| | TDGIA | 72.12±0.52 | 74.54±0.69 | 71.69±0.40 | 73.04±0.52 |
| | MetaGIA | 72.92±0.66 | 75.22±0.66 | 71.60±0.48 | 73.11±0.45 |
| Pubmed | *clean* | 89.93±0.27 | 89.96±0.25 | 88.10±0.33 | 88.93±0.18 |
| | PGD | 81.81±1.94 | 87.72±0.89 | 87.69±0.57 | 88.99±0.11 |
| | TDGIA | 86.62±1.05 | 88.86±0.40 | 87.55±0.60 | 88.80±0.12 |
| | MetaGIA | 87.58±0.75 | 88.23±0.93 | 87.40±0.62 | 88.78±0.16 |

Table 15: Node classification accuracy (%) on graph **Nettack targeted** attack in **transductive** learning.

| Dataset | Ptb-rate | HANG | HANG-GARNET | HANG-quad | HANG-quad-GARNET | GCN | GCN-GARNET |
|---|---|---|---|---|---|---|---|
| Cora | 1 | 75.54±3.10 | 82.41±0.80 | 76.99±3.16 | 83.01±0.84 | 70.06±0.81 | 79.75±2.35 |
| | 2 | 73.73±3.64 | 80.84±1.57 | 76.51±2.60 | 79.88±0.55 | 68.60±1.81 | 79.60±1.50 |
| | 3 | 68.43±4.23 | 80.48±1.69 | 73.13±2.85 | 79.76±0.72 | 65.04±3.31 | 74.42±2.06 |
| | 4 | 66.02±2.21 | 70.0±1.47 | 72.53±2.14 | 75.90±0.54 | 61.69±1.48 | 69.60±2.67 |
| | 5 | 60.12±3.63 | 67.83±1.87 | 68.80±2.55 | 69.28±1.34 | 55.66±1.95 | 67.04±2.05 |

# D  More Insights about Model Robustness

For a clearer insight into the energy concept within HANG, consider the "time" in graph neural flows as analogous to the "layers" in standard GNNs (note that ODE solvers basically discretize the "time", which indeed approximately turns the model into a layered one). Here, the feature vector $\mathbf{q}(t)$ and the momentum vector $\mathbf{p}(t)$ evolve with time, bound tightly by equation (10). Given that $\mathbf{q}(t)$ mirrors the node features at layer $t$, $\mathbf{p}(t)$ can be understood as the variation in node features over time - essentially, the evolution of node features between successive layers. Thus, our defined energy interweaves both the node feature and its rate of change across adjacent layers. The constant $H_{\text{net}}$ implies inherent constraints on the node features and their alteration pace over layers. Because $H_{\text{net}}$ processes the whole graph data and yields a scalar, it serves as a constraint on the global graph feature and its variation, which we opine to be crucial in countering adversarial attacks.

From an adversarial perspective, the attacker modifies either the node features or the underlying graph topology. These modifications are propagated through multiple aggregation steps, such as layers in conventional GNNs or integrals in graph ODEs. While the Hamiltonian considers the energy of the entire graph, adversarial attacks often target localized regions of the graph. The inherent global energy perspective of the Hamiltonian system makes it resilient to such localized attacks, as local perturbations often get "absorbed" or "mitigated" when viewed from the perspective of the entire system. When adversarial perturbations are introduced, they might indeed tweak the instantaneous features of certain nodes. However, the challenge lies in modifying the trajectory (or evolution) of these nodes (positions $p(t)$ and the variations $q(t)$) in the phase space in a manner that is aligned with the rest of the graph, all while upholding the energy conservation constraints. This feat is arduous,

if not impossible, without creating detectable inconsistencies elsewhere. This property ensures that the energy of each node feature is preserved over time and multiple aggregation steps. As a result, the distances between features of different nodes are preserved if their norms differ initially before the adversarial attack. The robustness of HANG against topology perturbation may stem from the fact that adversarial topology perturbation has small influence on the node feature energy. This can be seen from Fig. 1. When using HANG, the node feature before and after adversarial attack are relatively stable and well separated between nodes from different classes.

## E  Proof of Theorem 1

Recall that

$$\frac{\mathrm{d}\mathbf{X}(t)}{\mathrm{d}t} = \overline{\mathbf{A}}_G(\mathbf{X}(t))\mathbf{X}(t) = (\mathbf{A}_G(\mathbf{X}(t)) - \alpha\mathbf{I})\mathbf{X}(t), \tag{15}$$

Without loss of generality, we assume $\mathbf{X} \in \mathbb{R}^{|\mathcal{V}|\times 1}$ since the results can be generalized to $\mathbf{X} \in \mathbb{R}^{|\mathcal{V}|\times r}$ on a component-wise basis.

*Proof of 1).*

Given our assumptions where $\mathbf{A}_G(\mathbf{X}(t))$ is either column- or row-stochastic, the spectral radius of $\mathbf{A}_G(\mathbf{X}(t))$ is 1, as established by [42][Theorem 8.1.22]. This implies that the modulus of its eigenvalues is as most 1.

We define a Lyapunov function as $V(\mathbf{X}(t)) = \mathbf{X}(t)^\mathsf{T}\mathbf{X}(t) = \|\mathbf{X}(t)\|_2^2$ where $\|\cdot\|_2$ is the $\ell_2$ euclidean norm. Take the derivative of $V$ with respect to time, we have

$$\dot{V}(\mathbf{X}(t)) = \mathbf{X}^\mathsf{T}(t)\dot{\mathbf{X}}(t) + \dot{\mathbf{X}}^\mathsf{T}(t)\mathbf{X}(t) = \mathbf{X}^\mathsf{T}(t)\left(\overline{\mathbf{A}}_G(\mathbf{X}(t)) + \overline{\mathbf{A}}_G^\mathsf{T}(\mathbf{X}(t))\right)\mathbf{X}(t) \tag{16}$$

We next prove that $\dot{V}(\mathbf{X}(t)) \leq 0$ when $\mathbf{A}_G(\mathbf{X}(t))$ is a doubly stochastic attention matrix under the GRAND-nl setting.

$$\begin{aligned}
\dot{V}(\mathbf{X}(t)) &= \mathbf{X}^\mathsf{T}(t)\left(\overline{\mathbf{A}}_G(\mathbf{X}(t)) + \overline{\mathbf{A}}_G^\mathsf{T}(\mathbf{X}(t))\right)\mathbf{X}(t) \\
&= 2\mathbf{X}^\mathsf{T}(t)\overline{\mathbf{A}}_G(\mathbf{X}(t))\mathbf{X}(t) \\
&= 2\mathbf{X}^\mathsf{T}(t)\mathbf{A}_G(\mathbf{X}(t))\mathbf{X}(t) - 2\alpha\mathbf{X}^\mathsf{T}(t)\mathbf{X}(t)
\end{aligned} \tag{17}$$

Note that

$$|\mathbf{X}^\mathsf{T}(t)\mathbf{A}_G(\mathbf{X}(t))\mathbf{X}(t)| \leq \|\mathbf{X}\|_2\|\|\mathbf{A}_G(\mathbf{X}(t)\|\|_2\|\mathbf{X}\|_2 \tag{18}$$

where $\|\|\cdot\|\|_2$ is the spectral matrix norm induced by $\ell_2$ vector norm. The given inequality is a direct result of the Cauchy–Schwarz inequality and the properties of the induced norm. Drawing from [42][Theorem 5.6.9], we can currently establish that $\|\|\mathbf{A}_G(\mathbf{X}(t)\|\|_2 \geq 1$. Nevertheless, we intend to assert that this inequality is indeed an equality when $\mathbf{A}_G(\mathbf{X}(t))$ is doubly stochastic, i.e., $\|\|\mathbf{A}_G(\mathbf{X}(t)\|\|_2 = 1$. To see this, from Schur's bounds on the largest singular value [83][eq.(1)], we have

$$\|\|\mathbf{A}_G(\mathbf{X}(t)\|\|_2 \leq \sqrt{\|\|\mathbf{A}_G(\mathbf{X}(t)\|\|_\infty\|\|\mathbf{A}_G(\mathbf{X}(t)\|\|_1}$$

where $\|\|\cdot\|\|_1$ is the induced $\ell_1$ norm and $\|\|\cdot\|\|_\infty$ is the induced $\ell_\infty$ norm. Given that $\mathbf{A}_G(\mathbf{X}(t)$ is doubly stochastic, we observe that $\|\|\mathbf{A}_G(\mathbf{X}(t)\|\|_\infty = \|\|\mathbf{A}_G(\mathbf{X}(t)\|\|_1 = 1$. It now follows that $\|\|\mathbf{A}_G(\mathbf{X}(t)\|\|_2 \leq 1$, the equality is consequently confirmed. It follows from (18) that

$$|\mathbf{X}^\mathsf{T}(t)\mathbf{A}_G(\mathbf{X}(t))\mathbf{X}(t)| \leq \|\mathbf{X}\|_2^2 = \mathbf{X}^\mathsf{T}(t)\mathbf{X}(t) \tag{19}$$

From (17) and (19), it follows that

$$\dot{V}(\mathbf{X}(t)) \leq 2(1-\alpha)\|\mathbf{X}\|_2^2 \leq 0 \tag{20}$$

when $\alpha \geq 1$. The observation that $V(\mathbf{X}(t))$ does not increase over time ensures that $\mathbf{X}(t)$ stays bounded. In effect, we also have proved Lyapunov stability concerning the equilibrium point 0.

To prove asymptotic stability for the equilibrium point 0 when $\alpha > 1$, we need to show that the system not only remains bounded but also approaches 0 as $t \to \infty$. This is indicated by the fact that

when $\alpha > 1$, $\dot{V}(\mathbf{X}(t)) < 0$ unless $\mathbf{X}(t) = 0$. This signifies that $V(\mathbf{X}(t))$ strictly declines over time unless $\mathbf{X}(t) = 0$, thereby showing that the system will converge to 0 as $t \to \infty$. Notably, we can infer global asymptotic stability since $V$ is radially unbounded [84].

The proof for the GRAND-nl case is complete.

*Proof of 2).*

We next prove the claims under GRAND-l setting.

When $\mathbf{A}_G$ is either column- or row-stochastic, the spectral radius of $\mathbf{A}_G(\mathbf{X}(t))$ is 1, as established by [42][Theorem 8.1.22]. The modulus of its eigenvalues is as most 1. Consider the Jordan canonical form of $\mathbf{A}_G$ represented as $\mathbf{SJS}^{-1}$ where $\mathbf{J}$ stands as the Jordan form.

Given that our equation system now represents a linear time-invariant ODE, the solution to (15) can be expressed as:

$$\begin{aligned} \mathbf{X}(t) &= e^{\overline{\mathbf{A}}_G t}\mathbf{X}(0) \\ &= \mathbf{S}e^{\bar{\mathbf{J}}t}\mathbf{S}^{-1}\mathbf{X}(0) \end{aligned} \tag{21}$$

where $\bar{\mathbf{J}} = \mathbf{J} - \alpha\mathbf{I}$. Citing [42][3.2.2. page 177, 4-th equation], for $\alpha > 1$, BIBO and Lyapunov stabilities are attained. Notably, for $\alpha > 1$, the system achieves global asymptotic stability around the equilibrium point 0 under any disturbances.

Further, by considering strong connectedness and referencing the Perron-Frobenius theorem [42][Theorem 8.4.4], it is deduced that 1 serves as the simple eigenvalue for $\mathbf{A}_G$. In this case, when $\alpha = 1$, we still have BIBO stability and Lyapunov stability according to [42][3.2.2. page 177, 4-th equation].

*Proof of 3).*

We subsequently demonstrate that when $\mathbf{A}_G$ is selected to be column-stochastic and $\alpha = 1$, GRAND conserves a certain quantity interpretable as energy. Specifically, this "energy" refers to the sum of the elements of $\mathbf{X}(t)$, denoted as $\mathbf{1}^{\mathsf{T}}\mathbf{X}(t)$, with $\mathbf{1}$ representing an all-ones vector. From (15), this quantity remains conserved if

$$0 = \frac{\mathrm{d}\mathbf{1}^{\mathsf{T}}\mathbf{X}(t)}{\mathrm{d}t} = \mathbf{1}^{\mathsf{T}}\overline{\mathbf{A}}_G(\mathbf{X}(t))\mathbf{X}(t) \tag{22}$$

Given that $\mathbf{A}_G$ is column-stochastic, it follows directly that $\mathbf{1}^{\mathsf{T}}\overline{\mathbf{A}}_G = \mathbf{0}^{\mathsf{T}}$, validating (22).

Finally, we aim to prove that GRAND is asymptotic stable concerning a specified equilibrium vector when $\alpha = 1$ and the graph is aperiodic and strongly connected. Based on [42][Theorem 3.2.5.2.,Theorem 8.5.3.] and [85][Theorem 5.6.6], we observe that $\lim_{k\to\infty}(\mathbf{A}_G)^k = \lim_{k\to\infty}\mathbf{SJ}^k\mathbf{S}^{-1} = \mathbf{S}\mathbf{\Lambda}\mathbf{S}^{-1}$, where $\mathbf{\Lambda}$ is a diagonal matrix with the first element as 1 and all the others as 0:

$$\mathbf{\Lambda} = \begin{pmatrix} 1 & & & \\ & 0 & & \\ & & \ddots & \\ & & & 0 \end{pmatrix}$$

Since $\lim_{k\to\infty}(\mathbf{A}_G)^k$ maintains its column stochasticity and the rank of $\mathbf{S}\mathbf{\Lambda}\mathbf{S}^{-1}$ is 1, it follows that each column is the **same**. Due to the zeros in $\mathbf{\Lambda}$, the limit is the outer product of the first column of $\mathbf{S}$ and the first row of $\mathbf{S}^{-1}$. Since we have the same column for the outer product, we deduce that the first row of $\mathbf{S}^{-1}$ is $a\mathbf{1}^{\mathsf{T}}$ with $a$ being a scalar and $\mathbf{1}$ an all-ones vector. We have $\overline{\mathbf{A}}_G$ has an eigenvalue of 0, with the rest of the eigenvalues having *strictly negative real parts*. According to [42][3.2.2], it follows that

$$\lim_{t\to 0}\mathbf{X}(t) = \lim_{t\to 0}\mathbf{S}e^{\bar{\mathbf{J}}t}\mathbf{S}^{-1}\mathbf{X}(0) = \mathbf{S}\mathbf{\Lambda}\mathbf{S}^{-1}\mathbf{X}(0) \tag{23}$$

If $s := \mathbf{1}^{\mathsf{T}}\mathbf{X}(0)$ remains constant for any perturbed $\mathbf{X}(0)$, we have that $\lim_{t\to 0}\mathbf{X}(t) = sa\mathbf{s}$ where $\mathbf{s}$ is the first column of $\mathbf{S}$. We thus conclude that if the perturbation on $\mathbf{X}(0)$ maintains the column summations, i.e. the "energy", unchanged, the asymptotic convergence to the equilibrium vector $sa\mathbf{s}$ remains unaffected.

The proof is now complete.

# F    Proof of Theorem 2

The energy conservation of our system, represented as $H_{\text{net}}$ in (9), remains invariant over time. To understand this better, consider the following

$$
\begin{aligned}
\frac{\mathrm{d}H_{\text{net}}}{\mathrm{d}t} &= \sum_{i=1}^{n} \frac{\partial H_{\text{net}}}{\partial q_i} \dot{q}_i + \sum_{i=1}^{n} \frac{\partial H_{\text{net}}}{\partial p_i} \dot{p}_i \\
&= \sum_{i=1}^{n} \frac{\partial H_{\text{net}}}{\partial q_i} \frac{\partial H_{\text{net}}}{\partial p_i} + \sum_{i=1}^{n} \frac{\partial H_{\text{net}}}{\partial p_i} \left( -\frac{\partial H_{\text{net}}}{\partial q_i} \right) \\
&= 0.
\end{aligned}
$$

where the last equality follows from (10).

BIBO stability is inferred since, with an unbounded output, the energy conservation expressed by $H_{\text{net}}$ would be violated.

# G    Complete Algorithm Summary

We present the complete algorithm of HANG in Algorithm 1, which unfortunately had been delayed in its inclusion within the main paper due to space constraints.

## Limitations

While our work on graph neural flows presents promising advancements in enhancing adversarial robustness of GNNs using Hamiltonian-inspired neural ODEs, it is not without limitations. As we demonstrated in the paper, the notions of stability borrowed from dynamical systems, such as BIBO stability and Lyapunov stability, do not always guarantee adversarial robustness. Our finding that energy-conservative Hamiltonian graph flows improve robustness is only one facet of the broader landscape of potential stability measures. It is possible that other notions of stability, not covered in this work, could yield additional insights into adversarial robustness. Our current Hamiltonian graph neural flows do not explicitly account for quasi-periodic motions in the graph dynamics. The Kolmogorov-Arnold-Moser (KAM) theory, a foundational theory in Hamiltonian dynamics, is renowned for its analysis of persistence of quasi-periodic motions under small perturbations in Hamiltonian dynamical systems. While the energy-conserving nature of our Hamiltonian-inspired model inherently offers some level of robustness to perturbations, an explicit incorporation of KAM theory could potentially further improve the robustness, particularly in the face of quasi-periodic adversarial attacks. However, this is a complex task due to the high dimensionality of typical graph datasets and the intricacies involved in approximating quasi-periodic dynamics.

## Broader Impact

This research, centered on enhancing adversarial robustness in graph neural networks (GNNs), carries implications for various sectors, such as social media networks, sensor networks, and chemistry. By improving the resilience of GNNs, we can boost the reliability of AI-driven systems, contributing to greater efficiency, productivity, and cost-effectiveness. The shift towards automation may displace certain jobs, raising ethical concerns about income disparity and job security. Moreover, while our models enhance robustness, potential system failures can still occur, with impacts varying based on the application. Lastly, the robustness conferred might be exploited maliciously. Our work underscores the importance of diligent oversight, equitable technology implementation, and continuous innovation in the development of AI technologies.

---

**Algorithm 1:** Graph Node Embedding Learning with HANG

---

1  **Initialization:** Initialize the network modules including Hamiltonian function network $H_{\text{net}}$, the raw node features compressor network FC, and the final classifier.

2  **I. Training:**

3  **for** *Epoch* 1 *to* $N$ **do**

4     **1)** Perform the following to obtain the embedding $\mathbf{q}_k(T)$ for each node $k$:

5     **Input:** $\mathcal{G} = (\mathcal{V}, \mathcal{E})$ with raw node features

6     Apply FC to compress raw features for each node and get $\{(\mathbf{q}_k(0), \mathbf{p}_k(0))\}_{k=1}^{|\mathcal{V}|}$. Here, we divide the $2r$ dimensions into two equal segments. The first half acts as "position" vectors $\mathbf{q}_k(0) = (q_k^1, \ldots, q_k^r)$, while the second half acts as "momentum" vectors $\mathbf{p}_k(0) = (p_k^1, \ldots, p_k^r)$ that guide system evolution. For simplification, in code implementation, we may set the output feature dimension of the compressor FC as $r$ and designate $\mathbf{p}_k(0) = \mathbf{q}_k(0)$.

7     **2)** To better express the evolution dynamics mathematically, concatenate and relabel the node features as:

$$q(0) = \left(q^1(0), \ldots q^{r|\mathcal{V}|}(0)\right) = \left(\mathbf{q}_1(0), \ldots, \mathbf{q}_{|\mathcal{V}|}(0)\right).$$
$$p(0) = \left(p_1(0), \ldots p_{r|\mathcal{V}|}(0)\right) = \left(\mathbf{p}_1(0), \ldots, \mathbf{p}_{|\mathcal{V}|}(0)\right), \tag{24}$$

    Note that in the actual code implementation, the concatenation of node features is not necessary as $H_{\text{net}}$ is realized as either (12) or (13). The concatenate operation is only for mathematical formulation.

8     **3)** The trajectory of feature evolution is modelled as per the following canonical Hamilton's equations:

$$\dot{q}(t) = \frac{\partial H_{\text{net}}}{\partial p}, \quad \dot{p}(t) = -\frac{\partial H_{\text{net}}}{\partial q}, \tag{25}$$

    with the initial features $(q(0), p(0)) \in \mathbb{R}^{2r|\mathcal{V}|}$.

9     Various ODE solvers as provided by [18] and the symplectic-euler solver from [35] can be employed to solve (25). We refer readers to Appendix C.6 for more details.

10     **4)** Acquire the evolved features at time $T$ as $q(T)$, which is then decompressed into individual node features $\mathbf{q}_k(T)_{k=1}^{|\mathcal{V}|}$ for further utilization.

11     **5)** Utilize backpropagation to minimize the cross-entropy loss for node classification.

12     **6)** Perform validation over the validation split.

13     **7)** Save the model parameters.

14  **II. Testing:**

15  Load the model from the best validation epoch and perform **Step I.1-4).** to obtain the final feature embedding over the test split. Perform node classification.

---

