# OpenReview forum: "Adversarial Robustness in Graph Neural Networks: A Hamiltonian Approach"
_NeurIPS.cc/2023/Conference — NeurIPS 2023 spotlight_

### Official Review · Reviewer_Xukb · 2023-06-14

**Soundness:** 2 fair
**Presentation:** 1 poor
**Contribution:** 3 good
**Rating:** 7
**Confidence:** 5

**Summary:**

Drawing inspiration from physics principles, the paper proposes the use of conservative Hamiltonian neural flows to construct GNNs that are robust against adversarial attacks. The adversarial robustness of different neural flow GNNs is empirically evaluated on several benchmark datasets, considering a variety of adversarial attacks.

**Strengths:**

This paper investigates the vulnerability of graph neural networks (GNNs) to adversarial perturbations that affect both node features and graph topology. The study focuses on GNNs derived from various neural flows and explores their connection to different stability notions.

**Weaknesses:**

The followings are four weaknesses.

1. Paper's writing quality is poor due to excessive amount of equations, formulas, and its explanation in a view of physics. The authors should use the adversarial research language to bring out the main insight for why this work is important for adversarial research.

2. In addition, numerous words need to penetrate the core idea of equations rather than to revolve around technical terms.

3. The reviewer did not see the connection bridging the gap between Hamiltonian mechanics and energy conservation.

4. Is there any other simpler option that possess the notion of conservation other than Hamiltonian mechanics at all?  In addition, why is it important to conserve energy in a view of topological structure?

Overall, the reviewer recommends the authors to answer these questions using insightful language at least to the reviewer, not using mathematical or physical language. The reviewer does not want a theoretical proof, only want really contributional insight; Why is it important to consider energy or conservation and why are "energy and topology" bridged? Why is the hamiltonian important? Please answer using adversarial language

---

The authors' rebuttal has fully explained the connection between the energy conservation in the realm of the Hamiltonian and the adversarial robustness of GNNs using the adversarial research language. Thank the authors for their thorough commentary on the reviewer's point.

**Questions:**

Refer to Weaknesses

**Limitations:**

Refer to Weaknesses

---

> ### Author Rebuttal · Authors · 2023-08-09
>
> # Improve Paper Writing: Weakness 1 and 2
>
> We greatly value your feedback. Our paper delves into the relationship between the stability of graph ODE models and adversarial robustness. This connection is elaborated upon in segments such as Remark 2, lines 116-125, 165-180, and 280-285, where we correlate system stability with adversarial resistance.
>
> To enhance clarity:
>
> 1. Appendix Inclusion: Some secondary equations will be moved to an appendix, ensuring the main content remains focused.
>
> 2. Restrained Physics Emphasis: We will succinctly present physics-related sections, especially Section 4.1, while retaining essential insights. In-depth physics discussions will be directed to the appendix.
>
> 3. Highlighting Adversarial Context: We will provide more intuitive explanations for the choice of Hamiltonian mechanics and how it contributes to combatting adversarial attacks. Some example answers have been provided in our following responses, which will be further expanded in our revision.
>
>
>
> #  Hamiltonian Mechanics, Energy Conservation and Robustness: Weakness 3, 4 and 5
>
> Thank you for raising this point, and we are grateful for the opportunity to provide further clarity.
>
> Hamiltonian mechanics is not just a theoretical underpinning in our work but a core mechanism to underscore conservation principles. Specifically, the energy conservation of our system, represented as $H_{n e t}$ in equation (8), remains invariant over time, notably in the context of graph ODE models. To shed more light on this:
> \begin{align*}
> \frac{\mathrm{d} H\_{net}}{\mathrm{~d} t} & =\sum\_{i=1}\^n \frac{\partial H\_{net}}{\partial q\_i} \dot{q}\_i+\sum\_{i=1}\^n \frac{\partial H_{net}}{\partial p\_i} \dot{p}\_i\\\\
> & =\sum\_{i=1}\^n \frac{\partial H\_{net}}{\partial q\_i} \frac{\partial H\_{net}}{\partial p\_i}+\sum\_{i=1}\^n \frac{\partial H\_{net}}{\partial p\_i}\left(-\frac{\partial H\_{net}}{\partial q\_i}\right)\\\\
> & =0.
> \end{align*}
>
>
> What this means is that $H_{net}$ retains its value as $t$ changes, as stated by Theorem 2. Now, in the realm of physics, $H_{n e t}$ is interpreted differently across varied contexts, say as mechanical energy in a pendulum or as energy in an electron's charged field. In our framework, we model $H_{n e t}$ using a neural network model that ingests the full graph data and yields a scalar. This scalar, by design, remains invariant over time, echoing the conservation principle rooted in Hamiltonian mechanics.
>
>
> For a clearer insight into the energy concept within HANG, consider the "time" in graph neural flows as analogous to the "layers" in standard GNNs (note that ODE solvers basically discretize the "time", which indeed approximately turns the model into a layered one). Here, the feature vector $\mathbf{q}(t)$ and the momentum vector $\mathbf{p}(t)$ evolve with time, bound tightly by equation (9). Given that $\mathbf{q}(t)$ mirrors the node features at layer $t, \mathbf{p}(t)$ can be understood as the variation in node features over time - essentially, the evolution of node features between successive layers. Thus, our defined energy interweaves both the node feature and its rate of change across adjacent layers. The constant $H_{n e t}$ implies inherent constraints on the node features and their alteration pace over layers.
> Because $H_{net}$ processes the whole graph data and yields a scalar, it serves as a constraint on the global graph feature and its variation, which we opine to be crucial in countering adversarial attacks.
>
> Furthermore, by rewriting equation (9) in the following form,
> \begin{align}
> \left[\begin{array}{c}
> \dot{q}(t) \\\\
> \dot{p}(t)
> \end{array}\right]
> &=
> \mathbf{M}
> \left[\begin{array}{c}
>  \frac{\partial H_{\mathrm{net}}}{\partial p}\\\\
> \frac{\partial H_{\mathrm{net}}}{\partial q}
> \end{array}\right],
> \end{align}
> where $\mathbf{M}=\left[\begin{array}{cc}
> \mathbf{0} & \mathbf{I}\\\\
> -\mathbf{I} & \mathbf{0}
> \end{array}\right]$ is a real antisymmetric matrix and its eigenvalues (real parts) are all zeros,
> we can see that the antisymmetric matrix $\mathbf{M}$ leads to a rotation in feature space $\left[\begin{array}{c}
> q(t) \\
> p(t)
> \end{array}\right]$. This implies that the norm of a node remains constant over time while the phase of the feature may change. This property preserves distances between features of different nodes when their norms differ at the initial point. This, in turn, ensures a well-posed and robust forward propagation and learning process. Figure 1 in the main paper illustrates this property.
>
> From an adversarial perspective, the attacker modifies either the node features or the underlying graph topology. These modifications are propagated  through multiple aggregation steps, such as layers in conventional GNNs or integrals in graph ODEs.
> Here is where the Hamiltonian energy conservation property (as mentioned in our previous response) plays a crucial role. While the Hamiltonian considers the energy of the entire graph, adversarial attacks often target localized regions of the graph. The inherent global energy perspective of the Hamiltonian system makes it resilient to such localized attacks, as local perturbations often get "absorbed" or "mitigated" when viewed from the perspective of the entire system. When adversarial perturbations are introduced, they might indeed tweak the instantaneous features of certain nodes. However, the challenge lies in modifying the trajectory (or evolution) of these nodes (positions $p(t)$ and the variations $q(t)$) in the phase space in a manner that's aligned with the rest of the graph, all while upholding the energy conservation constraints. This feat is arduous, if not impossible, without creating detectable inconsistencies elsewhere.
> This property ensures that the energy of each node feature is preserved over time and multiple aggregation steps. As a result, the distances between features of different nodes are preserved if their norms differ initially before the adversarial attack.

---

> ### Author Response · Authors · 2023-08-11
> **Thanks to Reviewer Xukb**
>
> It's truly encouraging to know that our rebuttal addressed your concerns effectively! We genuinely value your insightful feedback and are grateful for your positive recognition of our work!

---

### Official Review · Reviewer_ehTk · 2023-07-02

**Soundness:** 3 good
**Presentation:** 3 good
**Contribution:** 2 fair
**Rating:** 7
**Confidence:** 4

**Summary:**

This paper proposes a robust GNN model by leveraging the notion of Hamiltonian Energy Conservation. Specifically, authors first analyze the stabilities and limitations of several neural ODE-based GNNs, which motivate the proposed model HANG that is inspired by Hamiltonian classical mechanics. Experimental results indicate that HANG outperforms prior ODE-based GNNs against various adversarial attacks on several realistic graph datasets.

**Strengths:**

- The paper is well written. All technical steps are easy to follow.
- Authors propose a novel approach to improve GNN robustness.
- Authors clearly motivate the proposed approach by analyzing the stability limitations of prior ODE-based GNNs.
- The proposed model has been evaluated against various adversarial attacks.

**Weaknesses:**

- The major issue of this work is that it only compares with a limited number of (and relatively weak) defense baselines. There are some recent methods such as [1, 2] showing better results than the baselines chosen in this paper. It would further improve the paper if authors could compare HANG with those methods.
- Figure 1 only shows the norm of node features become closer as time goes, which does not support authors' claim that node features gradually become similar (lines 182-183). Authors may want to show the norm of relative difference of node features is approaching to zero instead.

[1]: Geisler et al., “Robustness of Graph Neural Networks at Scale”, NeurIPS'21. \
[2]: Deng et al., “GARNET: Reduced-Rank Topology Learning for Robust and Scalable Graph Neural Networks”, LoG'22.

I'm willing to raise the score if my major concerns are addressed.

**Questions:**

- Is it possible to further improve GNN robustness by combining HANG with some prior defense methods (e.g. graph preprocessing-based methods), as they improve robustness in orthogonal ways?

**Limitations:**

Please refer to the Weaknesses section above for limitations.

---

> ### Author Rebuttal · Authors · 2023-08-09
>
> |Dataset|Attack|HANG|HANG-GUARD|HANG-quad|HANG-quad-GUARD|
> |-|-|-|-|-|-|
> |Cora|clean|87.13±0.86|86.54±0.57|79.68±0.62|81.23±0.70|
> ||PGD|78.37±1.84|86.23±0.55|79.05±0.42|80.91±0.67|
> ||TDGIA|79.76±0.99|85.56±0.34|79.54±0.65|81.11±0.76|
> ||MetaGIA|77.48±1.02|86.0±0.60|78.28±0.56|80.10±0.53|
> |Citeseer|clean|74.11±0.62|75.95±0.66|71.85±0.48|73.15±0.61|
> ||PGD|72.31±1.16|75.38±0.82|71.07±0.41|73.07±0.63|
> ||TDGIA|72.12±0.52|74.54±0.69|71.69±0.40|73.04±0.52|
> ||MetaGIA|72.92±0.66|75.22±0.66|71.60±0.48|73.11±0.45|
> |Pubmed|clean|89.93±0.27|89.96±0.25|88.10±0.33|88.93±0.18|
> ||PGD|81.81±1.94|87.72±0.89|87.69±0.57|88.99±0.11|
> ||TDGIA|86.62±1.05|88.86±0.40|87.55±0.60|88.80±0.12|
> ||MetaGIA|87.58±0.75|88.23±0.93|87.40±0.62|88.78±0.16|
>
> **Table R6.** Node classification accuracy (%) on graph **injection, evasion, non-targeted** attack in **inductive** learning.
>
> ----
>
> # New Baselines[1][2]: Weakness 1
>
> We appreciate the suggestion to broaden our comparison. The updated experiments are detailed subsequently.
>
> However, we wish to emphasize that our paper's primary focus is not solely on defense comparisons. Instead, we're keen on delving into the relationship between stability concepts and robustness **in the realm of graph neural ODE flows.** These stability concepts, intrinsic to graph neural flows, highlight the unique properties of such models. Therefore, while there are newer defense techniques like [1] and [2] with promising performance, our exploration into stability in graph neural flows presents a different facet, distinct from conventional defense methods.
>
> In response to the feedback, we've incorporated additional experiments in ``Table R7``, which include evaluations against defense techniques from [1] and [2].
>
> In our supplementary material Table S5,  we present the performance of our model when subject to Nettack. This result can be directly compared with the results from [2], as shown in ``Table R7``.
> Moreover, we also added experiments to demonstrate that our method is orthogonal to other defense methods. For example, the method presented in [2], GARNET, is a preprocessing defense mechanism that can be integrated with any GNN model. In our experiments, we employed the HANG model as the backbone and incorporated the GARNET defense. As depicted in ``Table R7``, the robust accuracy of HANG significantly benefits from the integration of GARNET. Furthermore, our combined approach, HANG-GARNET, outperforms the GCN-GARNET combination, underscoring the robustness and efficacy of our proposed model.
>
> The research by [1] introduced a robust aggregation GCN named Soft-Median-GCN. We evaluated this under two specific graph modification attacks, namely Nettack and Mettattack. The outcomes of these evaluations are in ``Table R7`` and ``Table R8``. It can be seen that our proposed models, HANG and HANG-quad, surpass Soft-Median-GCN under similar attack settings. This reaffirms the robustness of our models against such graph modification attacks.
>
> ----
> |Dataset|Ptb-rate|HANG|HANG-GARNET|HANG-quad|HANG-quad-GARNET|GCN|GCN-GATNET[2]|Soft-Median-GCN[1]|
> |-|-|-|-|-|-|-|-|-|
> |Cora|1|75.54±3.10|82.41±0.80|76.99±3.16|83.01±0.84|70.06±0.81|79.75±2.35|78.67±2.29|
> ||2|73.73±3.64|80.84±1.57|76.51±2.60|79.88±0.55|68.60±1.81|79.60±1.50|73.98±1.72|
> ||3|68.43±4.23|80.48±1.69|73.13±2.85|79.76±0.72|65.04±3.31|74.42±2.06|73.01±1.54|
> ||4|66.02±2.21|70.0±1.47|72.53±2.14|75.90±0.54|61.69±1.48|69.60±2.67|68.91±1.77|
> ||5|60.12±3.63|67.83±1.87|68.80±2.55|69.28±1.34|55.66±1.95|67.04±2.05|66.51±1.05|
>
> **Table R7.** Node classification accuracy (%) on graph **Nettack targeted** attack in **transductive** learning.
>
> ----
>
> # Visualization: Weakness 2
>
> Thank you for the valuable suggestion. In response to your suggestion, we have revised Figure 1 to provide a more accurate representation of the gradual similarity of node features. The new plot ``Figure R2`` in the uploaded pdf now displays the norm of the difference of node features, which indeed approaches zero as time progresses. This modification aligns with our claim and better conveys the intended message.
>
> # Combining HANG with Other Defenses: Question 1.
>
>  We would like to express our gratitude for your insightful suggestion. We recognize the potential benefits of combining defense mechanisms that operate through orthogonal strategies. To explore this, we integrated HANG with the GNNGuard preprocessing-based defense, known for its effectiveness in enhancing GNN robustness [P2] and GARNET, another preprocessing defense mechanism.
>
> We have conducted experiments to evaluate the combined performance of HANG with GNNGuard and GARNET, respectively. The results, as detailed in  ``Table R6 and Table R7``, demonstrate that this combination indeed improves the robustness accuracy of our model. This outcome underscores the compatibility of our approach with existing defense strategies and validates the potential for synergistic enhancements.
>
> Finally, it is also noteworthy that HANG can be combined with adversarial training techniques, such as PGD-AT, to further enhance its robustness. We refer the reviewer to the response **Adversarial Training (AT) and New AT Baselines** to **``Reviewer bDPL``** and ``Tables R1 and R2`` for more details.
>
> [P2] GNNGuard: Defending Graph Neural Networks against Adversarial Attacks, NeurIPS,2020.
>
>
> |Dataset|Ptb-rate(%)|HANG|HANG-quad|Soft-Median-GCN[1]|
> |-|-|-|-|-|
> |Polblogs|0|94.77±1.07|94.63±1.06|94.69±0.35|
> ||5|80.19±2.52|94.38±0.82|75.46±0.66|
> ||10|74.92±4.32|92.46±1.56|75.63±1.34|
> ||15|71.65±1.34|90.85±2.43|71.60±0.74|
> ||20|66.27±3.39|89.19±3.72|65.53±0.72|
> ||25|65.80±2.33|86.89±8.90|64.72±2.30|
> |Pubmed|0|85.08±0.20|85.23±0.14|85.22±0.11|
> ||5|85.08±0.18|85.12±0.18|84.17±0.08|
> ||10|85.17±0.23|85.05±0.19|81.90±0.21|
> ||15|85.0±0.22|85.15±0.17|79.05±0.51|
> ||20|85.20±0.19|85.03±0.19|75.74±0.23|
> ||25|85.06±0.17|84.99±0.16|72.75±0.39|
>
> **Table R8.** Node classification accuracy (%) on graph **Metattack non-targeted** attack in **transductive** learning.

---

> > ### Comment · Reviewer_ehTk · 2023-08-12
> > **Follow-up**
> >
> > Thanks for the detailed response and additional experiments, which clearly demonstrate the advantages of the proposed model. I would encourage authors to incorporate the clarification on the primary focus of this paper, as well as new empirical results of HANG+defense methods, in the revision, which would definitely improve the paper.
> >
> > Since authors have addressed my major concerns, I raise my score to 7.

---

> > > ### Author Response · Authors · 2023-08-12
> > > **Thanks to Reviewer ehTk**
> > >
> > > We are truly heartened by your feedback and are glad our rebuttal addressed your concerns! Your thoughtful observations will guide our revision. We commit to delineating the primary focus of the paper with greater clarity and will integrate the referenced experiments, along with the incorporation of other dense methods, into the revision.

---

### Official Review · Reviewer_Yk3z · 2023-07-07

**Soundness:** 3 good
**Presentation:** 3 good
**Contribution:** 2 fair
**Rating:** 5
**Confidence:** 1

**Summary:**

This paper explores the robustness of Graph Neural Networks (GNNs) against adversarial attacks. Drawing inspiration from principles in physics, the authors propose a novel model called Hamiltonian Neural Flows for constructing GNN models. The effectiveness of the proposed method is evaluated on various benchmark datasets.



**Strengths:**

The problem addressed in this paper is both significant and intriguing, and the proposed method is well-supported by principles from physics.

**Weaknesses:**

I am a non-expert in physics-based methods, my evaluation is based on my understanding of GNN methods and educated guesses. I suggest AC to ignore my evaluation if there are expert reviewers in the field.

Here are a couple of suggestions to improve the paper:

In my opinion, the comparison with existing baselines seems inadequate. The TDGIA paper was published in 2021, while MetaGIA was published in early 2022 or late 2021. It would be beneficial if the authors could include more recent graph attack papers and compare their proposed method against them.

Additionally, it would be interesting to see a comparison between the proposed method and the Graph Isomorphism Network (GIN).

**Questions:**

see cons above

**Limitations:**

Yes

---

> ### Author Rebuttal · Authors · 2023-08-09
>
> # Clarify and New Attacks:
>
> Thank you for your valuable feedback and suggestions. We truly value the time and effort you've dedicated to reviewing our paper.
> To provide some clarity, our work offers a fresh perspective on GNNs' adversarial robustness by probing the stability of graph ODE-based GNNs when viewed as dynamical systems. This insight underlines the critical role energy conservation plays in enhancing robustness. Drawing inspiration from Hamiltonian mechanics, our proposed HANG model paves a novel path to fortify GNNs against adversarial attacks. Importantly, our model can cooperate with other defense methods, such as graph preprocessing or adversarial training, to amplify its robustness. We are optimistic that this approach will spark further investigations in this field.
>
> Regarding your suggestion to include more recent graph attack papers for comparison, we agree that it would be beneficial to our study. New experimental results using the GANI(L) attack from a recent paper [W1] are provided in ``Table R4``. However, this attack appears less potent than the attack strategies we have explored in our paper. As evidenced in ``Table R4``, **HANG consistently exhibits better robustness,** even when faced with this newer attack approach.
>
> Additionally, ``Tables R1 and R2`` in our response to ``Reviewer bDLP`` and ``Tables R6 and R7`` in our feedback to ``Reviewer ehTk`` provide insights into our model's efficacy when combined with adversarial training and other pre-processing defense strategies, respectively.
>
>
> [W1]. Fang J, Wen H, Wu J, et al. GANI: Global Attacks on Graph Neural Networks via Imperceptible Node Injections[J]. arXiv preprint arXiv:2210.12598, 2022.
>
> | Dataset | Attack   | HANG         | HANG-quad    | GCN   | SGC   |
> |---------|----------|--------------|--------------|-------|-------|
> | Cora    | GANI(L)  | **77.48±0.92**   | 77.37±1.17   | 74.92 | 75.39 |
>
> **Table R4.** Node classification accuracy (%) on graph **injection, posioning, non-targeted GANI** attack in **transductive** learning.
>
> ---
>
> # Comparison with GIN:
>
> Thank you for your suggestion regarding the comparison with the Graph Isomorphism Network (GIN). We've taken your suggestion on board and incorporated GIN results into our evaluations, as shown in ``Table R5``. Notably, GIN, in line with other non-ODE-based GNNs, struggles under adversarial attacks. This result again highlights the unique resilience offered by HANG's energy conservation properties.
>
>
>
>
>
>
>
>
> | Dataset  | Attack   | HANG         | HANG-quad    | GIN         |
> |----------|----------|--------------|--------------|-------------|
> | Cora     | *clean*  | **87.13±0.86**   | 79.68±0.62   | 82.63±1.02  |
> |          | PGD      | 78.37±1.84   | **79.05±0.42**   | 32.86±0.23  |
> |          | TDGIA    | **79.76±0.99**   | 79.54±0.65   | 31.36±0.46  |
> |          | MetaGIA  | 77.48±1.02   | **78.28±0.56**   | 32.68±0.21  |
> | Citeseer | *clean*  | **74.11±0.62**   | 71.85±0.48   | 72.48±1.0   |
> |          | PGD      | **72.31±1.16**   | 71.07±0.41   | 28.20±3.37  |
> |          | TDGIA    | **72.12±0.52**   | 71.69±0.40   | 19.98±1.43  |
> |          | MetaGIA  | **72.92±0.66**   | 71.60±0.48   | 32.36±0.72  |
> | Pubmed   | *clean*  | **89.93±0.27**   | 88.10±0.33   | 86.46±0.24  |
> |          | PGD      | 81.81±1.94   | **87.69±0.57**   | 39.01±0.17  |
> |          | TDGIA    | 86.62±1.05   | **87.55±0.60**   | 40.35±1.13  |
> |          | MetaGIA  | **87.58±0.75**   | 87.40±0.62   | 40.47±0.41  |
>
> **Table R5.** Node classification accuracy (%) on graph **injection, evasion, non-targeted** attack in **inductive** learning.

---

### Official Review · Reviewer_bDPL · 2023-07-09

**Soundness:** 2 fair
**Presentation:** 2 fair
**Contribution:** 2 fair
**Rating:** 6
**Confidence:** 2

**Summary:**

Since neural ordinary differential equation networks can show inherent robustness, in this work the authors try to perform an extensive study on different graph neural flows along with their stability on different stability notions like BIBO stability, Lyapunov stability, Structural stability and Conservative stability. The authors find that the graph neural flows using hamiltonian energy functions can achieve improved empirical adversarial robustness on using black box attacks like PGD and TDGIA. The authors show that designing graph neural flows which can ensure conservation stability along with Lyapunov stability can help in achieving improved adversarial robustness. As common in literature the authors generate a white box attack from a surrogate model and then transfer it to the black box model to testify the robustness of the black box model. Evaluation is done on node injection attacks as well as graph manipulation attacks, and significantly improved robustness is seen in both cases.

**Strengths:**

* I think using graph neural flows to understand if we can achieve inherent robustness without the need for adversarial training is interesting.
* The way the authors have related different stability criteria with robustness is quite interesting.
* The results show significant improvements.


**Weaknesses:**

* There have been many instances where defences claiming to be robust have been later evaded using adaptive attacks [1]. Adversarial training has been the most successful defence strategy and the current successful defences use adversarial training to achieve robustness. Therefore, it is necessary that the robustness of the proposed method is evaluated properly. I think that the surrogate model-based black box attacks are not strong enough to get the worst-case robustness. Therefore, it is important that the evaluation on strong white-box attacks like PGD, PGD with max-margin loss / Carlini and Wagner attack [2] is used to testify robustness, and this evaluation should be carried out in a white-box setting. Further, it is important that the authors share a robust accuracy (both white box and black box) vs constraint threat model plot for the proposed approach, GAT and GAT trained using PGD-AT [3].

* Comparison is done only with models trained using standard training. It is important to include adversarially trained models as baselines. For instance, the authors should include a PGD-AT trained model as a baseline in all the tables.

* In the case of GNNs it is important to ensure that the attack graph remains imperceptible and the attacked nodes in the case of node injection attacks cannot be pruned off. Therefore other than robust accuracy imperceptibility of the graph for a given threat model is also a very important metric to be considered. Similar to [], I request the authors to also include a comparison on the imperceptibility of the attack on the proposed Graph neural flow network and the baselines like GAT. It is important to ensure that the imperceptiblity of the proposed defence is either lower or equal to the baselines against the PGD and TDGIA attacks.

* The authors should also look at these works [4,5] and try to compare their methods with them .

[1] Athalye, Anish et al. “Obfuscated Gradients Give a False Sense of Security: Circumventing Defenses to Adversarial Examples.” International Conference on Machine Learning (2018).
[2] Carlini, Nicholas, and David Wagner. "Towards evaluating the robustness of neural networks."
[3] Madry, Aleksander et al. “Towards Deep Learning Models Resistant to Adversarial Attacks.”
[4] Li, Jintang et al. “Spectral Adversarial Training for Robust Graph Neural Network.”
[5] Li, Jintang et al. “Spectral Adversarial Training for Robust Graph Neural Network.”


**Questions:**

t would be great if the authors can address the raised concerns in the weakness section.

**Limitations:**

Yes the authors seem to have addressed the limitations. I would suggest the authors to kindly look at the weakness section for suggestions and comments.

---

> ### Author Rebuttal · Authors · 2023-08-09
>
> # White Box Attack: Weakness 1
>
> We are grateful for your attention to the robustness evaluation. We **do include the results of the white-box attack in Table S3 of our supplementary material.** The results clearly demonstrate that both HANG and HANG-quad exhibit superior robustness compared to the other baseline models. We thank you for highlighting this, and we will ensure that these results are appropriately emphasized in the main paper as well.
>
> # Adversarial Training (AT) and New AT Baselines: Weakness 2, 3 and 5
>
>
> Thank you for your suggestion. We agree that comparing our model with adversarially trained models would provide a more comprehensive evaluation of its performance. However, we would like to emphasize the core objective of our paper is to elucidate the inherent stability and robustness of graph-ODE-based models.
>  It is also noteworthy that our HANG model can be combined with adversarial training techniques, such as PGD-AT, to further enhance its robustness.
>
>
> To address the reviewer's concern, we carried out more experiments applying **PGD-AT** to both the GAT and HANG models and assessed their robust accuracy under PGD, TDGIA, and METAGIA attacks, encompassing both white-box and black-box scenarios.
> As depicted in ``Tables R1 and R2``, PGD-AT does provide an uptick in the robustness of all models. Yet, ``Table R2`` interestingly reveals that the effect of PGD-AT on the GAT model's resilience to the **distinct white-box METAGIA attack** is somewhat limited. This observation underscores the boundaries of PGD-AT's effectiveness, especially when attackers adopt tactics not synonymous with AT techniques.
>
> On the other hand, HANG consistently achieves commendable performance across the spectrum of attacks, with a marked improvement when integrated with PGD-AT. It's noteworthy that  HANG-AT  largely withstands the onslaught of these attacks, a testament to HANG's inherent robustness.
>
> We would also like to point out that the new baseline [4,5], referencing the same publication, delves into adversarial training from a graph spectral viewpoint. We incorporated the Spectral Adversarial Training (SAT) from [4,5] into our HANG paradigm, giving rise to the HANG-SAT variant. As shown in ``Tables R1 and R2``, the fusion of SAT notably elevates the robustness of HANG against diverse attacks, with minimal post-attack accuracy reduction, further demonstrating HANG's innate resilience.
>
> Furthermore, we have taken the initiative to augment our experimentation by including comparisons with recently proposed defense methods. For further details, please refer to ``Table R6, Table R7, and Table R8.``
>
>
> # Imperceptibility: Weakness 4
>
> Thank you for your insightful comments regarding the imperceptibility of attacks on GNNs.  We understand the importance of this facet in adversarial studies and value the chance to elaborate on its role within our HANG framework.
>
> 1. Firstly, it is important to emphasize that the nature of graph imperceptibility largely stems from the attack methodology, as highlighted in references [P1], [P2], and [P3], rather than the targeted model itself. Be it under white-box or black-box conditions, the subtlety of the graph structure post-attack remains consistent, regardless of the model in focus. Were there a new measure of imperceptibility, the research community would likely craft an attack to ensure its adherence.
> In our work, the TDGIA, Metattack, and Nettack techniques utilized have built-in measures to ensure imperceptibility to the graph. These measures strictly adhere to the prescribed attack budgets. For a deeper dive into this aspect, we direct your attention to Table S4 in the supplementary material, which showcases experimental results across different attack intensities.
> Furthermore, it's pertinent to mention that the design philosophy behind HANG is distinct from defenses that either detect or make alterations to the graph structure.
>
> 2. Secondly, we initiated an examination of unnoticeability by conducting a degree distribution test under various threat models. The results, presented in ``Figure R1`` of the attached document, unveil a high degree of structural similarity between the original and modified graph topologies, affirming the imperceptibility we aimed to achieve.
>
>
> Additionally, reference [P1] sheds light on another advanced imperceptibility metric termed the homophily distribution. Here, homophily denotes the propensity of nodes to establish connections with peers sharing similar attributes or labels (Equation (6) in [P1]).
> To provide more clarity, ``Figure R3`` in the uploaded pdf presents a comparison of the homophily distributions before and after the attacks on both HANG and the baseline models.
> To rigorously evaluate our approach's resilience, we also incorporated the homophily constraint into the attacks using Harmonious Adversarial Objective (HAO)[P1]. This adaptation has resulted in a notably consistent homophily distribution, evident in the second row of ``Figure R3``. The results in ``Table R3``  demonstrate the enhanced performance of our proposed model over the baselines, emphasizing its robustness even with increased imperceptibility constraints.
>
> [P1] Yongqiang Chen, Han Yang, Yonggang Zhang, Kaili Ma, Tongliang Liu, Bo Han, James Cheng, "Understanding and Improving Graph Injection Attack by Promoting Unnoticeability," ICLR, 2022.
>
> [P2] Fang J, Wen H, Wu J, et al. GANI: Global Attacks on Graph Neural Networks via Imperceptible Node Injections[J]. arXiv preprint arXiv:2210.12598, 2022.
>
> [P3] Liu Z, Wang G, Luo Y, et al. What Does the Gradient Tell When Attacking the Graph Structure[J]. arXiv preprint arXiv:2208.12815, 2022.
>
>
> ## _Table R1, Table R2, and Table R3 are provided in the top global rebuttal window._

---

> > ### Comment · Reviewer_bDPL · 2023-08-12
> > **Reply to Authors**
> >
> > I am not sure if the authors have evaluated their method properly on the adaptive attacks. Other answers given by the authors seem to satisfy my concerns and therefore, I would like to raise my score to 6.

---

> > > ### Author Response · Authors · 2023-08-17
> > >
> > > Apologies for the delayed response.
> > >
> > > Thank you for your thoughtful feedback. We appreciate your comments, which have improved our paper.
> > >
> > > In our exploration of potential gradient masking, we observed that the accuracy under white-box attacks in Table S3 is lower than under black-box attacks in Table 2. This observation suggests the absence of significant gradient masking within our model. Additionally, as indicated in Table S4, the attack success rate increases consistently with larger distortion, correlating with the gradual decrease in accuracy. These empirical findings, in alignment with the insights provided in section 3.1 of the paper [1] you referenced, indicate that our model may not exhibit pronounced gradient masking.
> > >
> > >  We acknowledge the importance of this investigation and recognize that more exploration on this topic is warranted for future work.
> > >
> > > Thank you once again for your invaluable feedback and insights. Your engagement greatly enriches our work and contributes to its refinement.

---

### Author Rebuttal · Authors · 2023-08-09

In response to the reviewers' feedback, we have undertaken several substantial efforts to enhance the quality and comprehensibility of our paper. Here is a summary of the key actions we have taken:

1. **New Experiments**: We have executed a series of new experiments to bolster our findings and comparisons:
    - Conducted adversarial training on our model, presenting the results in ``Table R1`` and ``Table R2``.
    - Introduced new baselines and defense methods, capturing the outcomes in ``Tables R5, R7 and R8``.
    - Explored the implications of a novel graph attack method, reflected in ``Table R4``.
    - Explored the synergy of preprocessing defence methods combined with our model, showcased in ``Tables R6 and R7``.
    - Implemented more imperceptible attacks ``Table R3`` and visualized their imperceptibility through ``Figures R1 and R3``.
2. **Energy Conservation Visualization**: To underline the energy conservation property inherent to our approach, we introduced a visualization in ``Figure R2`` depicting the norm of feature differences versus time. This illustration effectively highlights the consistent energy preservation in our model.
3. **Intuitive Explanation of Hamiltonian and Energy Conservation**: Recognizing the need for clarity, we have provided a more intuitive explanation regarding the significance of Hamiltonian mechanics and how energy conservation intricately links to adversarial robustness. These insights aim to enhance the reader's understanding of our approach.
4. **Core Contribution Clarification**: We have further clarified and succinctly articulated the core contribution of our paper. This ensures that the reader grasps the pivotal insights and implications of our research.



### New experimental results: **Tables R1 - R3**:

|Dataset|Attack|HANG|HANG-AT|GAT|GAT-AT|HANG-SAT[4][5]|
|-|-|-|-|-|-|-|
|Pubmed|*clean*|**89.93±0.27**|89.45±0.33|87.41±1.73|85.88±0.47|88.38±0.48|
||PGD|81.81±1.94|**89.03±0.19**|48.94±12.99|77.78±8.39|87.65±1.77|
||TDGIA|86.62±1.05|87.16±3.89|47.56±3.11|79.56±5.76|**87.55±2.11**|
||MetaGIA|87.58±0.75|**88.93±0.18**|44.75±2.53|81.86±3.44|88.22±0.22|



**Table R1.** Node classification accuracy (%) on graph **injection, evasion, non-targeted, black-box** attack in **inductive** learning.

----


|Dataset|Attack|HANG|HANG-AT|GAT|GAT-AT|HANG-SAT[4][5]|
|-|-|-|-|-|-|-|
|Pubmed|*clean*|**89.93±0.27**|89.42±0.22|87.41±1.73|85.90±0.44|88.38±0.48|
||PGD|68.62±2.82|**88.86±0.15**|38.04±4.91|80.85±4.51|88.19±0.28|
||TDGIA|69.56±3.16|**88.98±0.20**|24.43±4.10|82.16±4.10|88.28±0.34|
||MetaGIA|84.64±1.20|**88.61±0.20**|40.02±1.34|44.37±6.48|85.76±4.12|


**Table R2.** Node classification accuracy (%) on graph **injection, evasion, non-targeted, white-box** attack in **inductive** learning.

----

| Dataset | Attack | HANG | HANG-quad | GRAND | GAT | GCN |
|---|---|---|---|---|---|---|
| Cora | *clean* | 86.32±0.53 | 80.56±0.65 | 87.62±0.75 | 87.47±0.51 | **87.99±0.45** |
| | TDGIA+HAO | **82.81±0.96** | 80.60±0.65 | 42.21±4.21 | 50.04±15.04 | 35.19±0.96 |
| | MetaGIA+HAO | **81.30±0.74** | 80.05±0.87 | 44.33±1.70 | 41.58±2.83 | 39.95±0.41 |
| Citeseer |*clean*| 74.78±0.71 | 73.22±0.74 | **76.31±0.86** | 72.45±0.89 | 73.22±0.65 |
| | TDGIA+HAO | **73.38±0.35** | 72.38±0.48 | 36.79±2.98 | 31.18±8.62 | 24.41±2.32 |
| | MetaGIA+HAO | 70.05±0.56 | **72.37±0.83** | 20.57±3.59 | 18.11±1.82 | 18.41±0.33 |
| Pubmed |*clean*| **90.04±0.24** | 89.02±0.20 | 88.66±0.17 | 87.44±0.19 | 88.24±0.23 |
| | TDGIA+HAO | 86.19±1.45 | **88.89±0.17** | 42.45±1.20 | 44.35±3.11 | 43.56±1.03 |
| | MetaGIA+HAO | 85.89±0.87 | **88.87±0.79** | 57.49±0.99 | 39.57±0.55 | 39.58±0.05 |



**Table R3.** Node classification accuracy (%) on graph **injection, evasion, non-targeted, black-box** attack in **inductive** learning.

---

### Decision · Program_Chairs · 2023-09-21

**Decision:**

Accept (spotlight)

**Comment:**

The reviewers consistently think the paper studies an important research problem on the robustness of GNN. I therefore recommend acceptance of the paper. Please revise the paper according to the reviews in the camera-ready version.